# DEFORMABLE GRAPH TRANSFORMER

## ABSTRACT

Transformer-based models have recently shown success in representation learning on graph-structured data beyond natural language processing and computer vision. However, the success is limited to small-scale graphs due to the drawbacks of full dot-product attention on graphs such as the *quadratic* complexity with respect to the number of nodes and message aggregation from enormous *irrelevant* nodes. To address these issues, we propose Deformable Graph Transformer (DGT) that performs sparse attention via dynamically selected *relevant* nodes for efficiently handling large-scale graphs with a *linear* complexity in the number of nodes. Specifically, our framework first constructs multiple node sequences with various criteria to consider both structural and semantic proximity. Then, combining with our learnable Katz Positional Encodings, the sparse attention is applied to the node sequences for learning node representations with a significantly reduced computational cost. Extensive experiments demonstrate that our DGT achieves superior performance on 7 graph benchmark datasets with 2.5 ∼ 449 times less computational cost compared to transformer-based graph models with full attention.

## 1 INTRODUCTION

Transformer (Vaswani et al., 2017) has proven its effectiveness in modeling sequential data in various tasks such as natural language understanding (Devlin et al., 2019; Yang et al., 2019; Brown et al., 2020) and speech recognition (Zhang et al., 2020; Gulati et al., 2020). Beyond sequential data, recent works (Dosovitskiy et al., 2021; Liu et al., 2021; Yang et al., 2021; Carion et al., 2020; Zhu et al., 2021; Zhao et al., 2021) have successfully generalized Transformer to various computer vision tasks such as image classification (Dosovitskiy et al., 2021; Liu et al., 2021; Yang et al., 2021), object detection (Carion et al., 2020; Zhu et al., 2021; Song et al., 2021), and 3D shape classification (Zhao et al., 2021). Inspired by the success of Transformer-based models, there have been recent efforts to apply the Transformer to graph domains by using graph structural information through structural encodings (Ying et al., 2021; Dwivedi & Bresson, 2020; Mialon et al., 2021; Kreuzer et al., 2021), and they have achieved the best performance on various graph-related tasks.

However, most existing Transformer-based graph models have difficulty in learning representations on large-scale graphs while they have shown their superiority on small-scale graphs. Since the Transformer-based graph models perform self-attention by treating each input node as an input token, the computational cost is quadratic in the number of input nodes, which is problematic on large-scale graphs. In addition, different from graph neural networks that aggregate messages from local neighborhoods, Transformer-based graph models globally aggregate messages from numerous nodes. So, on large-scale graphs, a huge number of messages from falsely correlated nodes often overwhelm the information from relevant nodes. As a result, Transformer-based graph models often exhibit poor generalization performance. A simple method to address these issues is performing masked attention where the key and value pairs are restricted to neighborhoods of query nodes (Dwivedi & Bresson, 2020). But, since the masked attention has a fixed small receptive field, it struggles to learn representations on large-scale graphs that require a large receptive field.

In this paper, we propose a novel Transformer for graphs named Deformable Graph Transformer (DGT) that performs sparse attention with a small set of key and value pairs adaptively selected considering both semantic and structural proximity. To be specific, our approach first generates multiple node sequences for each query node with diverse sorting criteria such as Personalized PageRank (PPR) score, BFS, and feature similarity. Then, our Deformable Graph Attention (DGA),

a key module of DGT, dynamically adjusts offsets to choose the key and value pairs on the generated node sequences and learns representations with selected key-value pairs. In addition, we present simple and effective positional encodings to capture structural information. Motivated by Katz index (Katz, 1953), which is used for measuring connectivity between nodes, we design Katz Positional Encoding (Katz PE) to incorporate structural similarity and distance between nodes on a graph into the attention. Our extensive experiments show that DGT achieved good performances on 7 benchmark datasets and outperformed existing Transformer-based graph models on all 8 datasets at a significantly reduced computational cost.

Our **contributions** are as follows: **(1)** We propose Deformable Graph Transformer (DGT) that performs sparse attention with a reduced number of keys and values for learning node representations, which significantly improves the scalability and expressive power of Transformer-based graph models. **(2)** We design deformable attention for graph-structured data, Deformable Graph Attention (DGA), that flexibly attends to a small set of relevant nodes based on various types of the proximity between nodes. **(3)** We present learnable positional encodings named Katz PE to improve the expressive power of Transformer-based graph models by incorporating structural similarity and distance between nodes based on Katz index (Katz, 1953). **(4)** We validate the effectiveness of the Deformable Graph Transformer with extensive experimental results that our DGT achieves the best performance on 7 graph benchmark datasets with $2.5 \sim 449$ times less computational cost compared to transformer-based graph models with full attention.

## 2 RELATED WORKS

**Graph Neural Networks.** Graph Neural Networks have become the *de facto* standard approaches on various graph-related tasks (Kipf & Welling, 2017; Hamilton et al., 2017; Wu et al., 2019; Xu et al., 2018; Gilmer et al., 2017). There have been several works that apply attention mechanisms to graph neural networks (Rong et al., 2020; Veličković et al., 2018; Brody et al., 2022; Kim & Oh, 2021) motivated by the success of the attention. GAT (Veličković et al., 2018) and GATv2 (Brody et al., 2022) adaptively aggregate messages from neighborhoods with the attention scheme. However, the previous works often show poor performance on heterophilic graphs due to their homophily assumption that nodes within a small neighborhood have similar attributes and potentially the same labels. So, recent works (Abu-El-Haija et al., 2019; Pei et al., 2020; Zhu et al., 2020; Park et al., 2022) have been proposed to extended message aggregation beyond a few-hop neighborhood to cope with both *homophilic* and *heterophilic* graphs. H2GCN (Zhu et al., 2020) separates input features and aggregated features to preserve information of input features. Deformable GCN (Park et al., 2022) improves the flexibility of convolution by performing deformable convolution.

**Transformer-based Graph Models.** Recently, (Ying et al., 2021; Dwivedi & Bresson, 2020; Mialon et al., 2021; Kreuzer et al., 2021; Wu et al., 2021) have adopted the Transformer architecture for learning on graphs. Graphormer (Ying et al., 2021) and GT (Dwivedi & Bresson, 2020) are built upon the standard Transformer architectures by incorporating structural information of graphs into the dot-product self-attention. However, these approaches, which we will refer to as 'graph Transformers' for brevity, are not suitable for large-scale graphs. It is because referencing numerous key nodes for each query node is prohibitively costly, and that hinders the attention module from learning the proper function due to the noisy features from irrelevant nodes. Although restricting the attention scope to local neighbors is a simple remedy to reduce the computational complexity, it leads to a failure in capturing local-range dependency, which is crucial for large-scale or heterophilic graphs. To mitigate the shortcomings of existing Transformer-based graph models, we propose DGT equipped with deformable sparse attention that dynamically selected relevant nodes to efficiently learn powerful representations on both homophilic and heterophilic graphs with significantly improved scalability.

**Sparse Transformers in Other Domains.** Transformer (Vaswani et al., 2017) and its variants have achieved performance improvements in various domains such as natural language processing (Devlin et al., 2019; Brown et al., 2020) and computer vision (Dosovitskiy et al., 2021; Carion et al., 2020). However, these models require quadratic space and time complexity, which is especially problematic with long input sequences. Recent works (Choromanski et al., 2021; Jaegle et al., 2021; Kitaev et al., 2020) have studied this issue and proposed various efficient Transformer architectures. (Choromanski et al., 2021; Xiong et al., 2021) study the low-rank approximation for attention to

reduce the complexity. Perceiver (Jaegle et al., 2021; 2022) leverages a cross-attention mechanism to iteratively distill inputs into latent vectors to scale linearly with the input size. Sparse Transformer (Child et al., 2019) uses pre-defined sparse attention patterns on keys by restricting the attention pattern to be fixed local windows. (Zhu et al., 2021) also proposes sparse attention that dynamically samples a set of key/value pairs for each query without a fixed attention pattern. Inspired by the deformable attention (Zhu et al., 2021), we propose deformable attention for graph-structured data that flexibly attends to informative key nodes considering various types of the proximity between nodes via multiple node sequences and our learnable Katz Positional Encodings.

## 3 METHODS

The goal of our architecture is to address the limitations of Transformer-based graph models and generalize Transformers on large-scale graphs. Specifically, existing Transformer-based graph models suffer from multiple challenges: 1) a scalability issue caused by the quadratic computational cost in regards to the number of nodes and 2) aggregation of distracting information since an enormous number of nodes are aggregated. To address the challenges, we propose Deformable Graph Transformer (DGT). Our framework is composed of two main components: 1) deformable attention that attends to only a small set of adaptively selected key nodes considering diverse relations between nodes, and 2) positional encoding that captures structural similarity and distance between nodes. Before introducing our proposed architectures, we revisit the basic concepts of graph neural networks and attention in Transformers.

### 3.1 PRELIMINARIES

**Graph Neural Networks (GNNs).** Consider an undirected graph $\mathcal{G} = (\mathcal{V}, \mathcal{E})$ with a set of $N$ nodes $\mathcal{V} = \{v_1, v_2, \ldots, v_N\}$ and a set of edges $\mathcal{E} = \{(v_i, v_j) \mid v_i, v_j \in \mathcal{V}\}$ where the nodes $v_i, v_j \in \mathcal{V}$ are connected. Each node $v_i \in \mathcal{V}$ has a feature vector $\mathbf{x}_i \in \mathbb{R}^F$, where $F$ is the dimensionality of the node feature, and a set of neighborhoods $\mathcal{N}(i) = \{v_j \in \mathcal{V} | (v_i, v_j) \in \mathcal{E}\}$.

Given a graph $\mathcal{G}$ and a set of node features $\{\mathbf{x}_i\}_{i=1}^N$, Graph Neural Networks (GNNs) aim to learn each node representation by an iterative aggregation of transformed representations of the node itself and its neighborhoods as follows:

$$\mathbf{h}_i^{(l)} = \sigma \left( \mathbf{W}^{(l)} \left( c_{ii}^{(l)} \mathbf{h}_i^{(l-1)} + \sum_{v_j \in \mathcal{N}(i)} c_{ij}^{(l)} \mathbf{h}_j^{(l-1)} \right) \right), \tag{1}$$

where $\mathbf{h}_i^{(l)} \in \mathbb{R}^{d^{(l)}}$ is a hidden representation of node $v_i$ in the $l$-th GNN layer, $\mathbf{h}_i^{(0)} = \mathbf{x}_i$, $\mathbf{W}^{(l)} \in \mathbb{R}^{d^{(l)} \times d^{(l-1)}}$ is a learnable weight matrix at the $l$-th GNN layer, $\sigma$ is a non-linear activation function, $c_{ij}^{(l)}$ and $c_{ii}^{(l)}$ represent weights for aggregation characterized by each GNN. For example, GCN Kipf & Welling (2017) can be represented as a form of (1) if $c_{ij} = (\deg(i)\deg(j))^{-1/2}$ and $c_{ii} = (\deg(i))^{-1}$, where $\deg(i)$ is the degree of node $v_i$, and GAT Veličković et al. (2018) learns $c_{ij}^{(l)}$ and $c_{ii}^{(l)}$ based on the attention mechanism.

**Attention in Transformer.** Transformers Vaswani et al. (2017) have shown their superior performance based on multi-head self attention mechanisms. Given a query index $q \in \Omega_q$ with a corresponding vector $\mathbf{z}_q \in \mathbb{R}^c$ and a set of key/value vectors $\mathcal{F} = \{\mathbf{f}_k\}_{k \in \Omega_k}$, where $\Omega_q, \Omega_k$ are the set of query and key indices, the Multi-Head Attention (MHA) is formulated as follows:

$$\text{MHA}(\mathbf{z}_q, \mathcal{F}) = \sum_{m=1}^M \mathbf{W}_m \left[ \sum_{k \in \Omega_k} \mathbf{A}_{mqk} \cdot \mathbf{W}'_m \mathbf{f}_k \right], \tag{2}$$

where $m$ is the index of $M$ attention heads, $\mathbf{W}_m \in \mathbb{R}^{c \times c_v}$ and $\mathbf{W}'_m \in \mathbb{R}^{c_v \times c}$ are weight matrix parameters. The attention weight $\mathbf{A}_{mqk}$ between the query $\mathbf{z}_q \in \mathbb{R}^c$ and the key $\mathbf{f}_k \in \mathbb{R}^c$, which is generally calculated as $\mathbf{A}_{mqk} = \frac{\exp[\mathbf{z}_q^\top \mathbf{U}_m^\top \mathbf{V}_m \mathbf{f}_k / \sqrt{c_v}]}{Z}$, where $Z \in \mathbb{R}$ is a normalization factor to achieve $\sum_{k \in \Omega_k} \mathbf{A}_{mqk} = 1$ and $\mathbf{U}_m, \mathbf{V}_m$ are weight matrices to compute queries and keys, respectively. Although the multi-head attention operation contributes to the success of the Transformer-based models, it has a known issue of substantial computational costs and large memory.

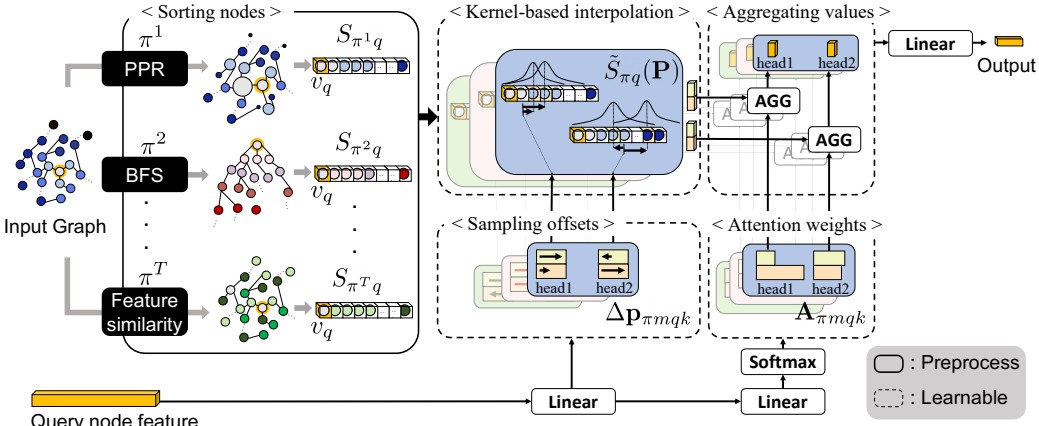

Figure 1: Overview of the deformable graph attention module. In the pre-processing phase, NodeSort module first constructs multiple node sequences $\{S_{\pi q}\}_{\pi \in \Pi}$ depending on query node $v_q$ by sorting nodes through diverse criteria $\pi \in \Pi$. Then, the kernel-based interpolation is applied on each offset to get values, whose offsets are computed by the queries with a linear projection. The deformable graph attention module aggregates the values of each head with attention weights to generate output.

This scalability issue has been extensively studied in the literature. However, most efficient attention methods have been developed by either limiting the attention within a local neighborhood or approximating an attention map with a low-rank matrix. The former approaches are problematic when neighbors belong to different classes as heterophilic graphs, and the latter approaches tend to yield *dense* attention leading to enormous messages from irrelevant nodes. So, we adopt a more flexible and efficient Deformable Attention Zhu et al. (2021); Xia et al. (2022) that performs a *sparse* attention by dynamically sampling a small set of relevant keys without a predefined window. Let $\mathbf{f} \in \mathbb{R}^{c \times h \times w}$ be an input feature map and $\mathbf{p}_q$ be a position of query $\mathbf{z}_q$. Then, the DeFormable Attention (DFA) is defined as

$$\mathrm{DFA}\left(\mathbf{z}_q, \mathbf{p}_q, \mathcal{F}\right) = \sum_{m=1}^{M} \mathbf{W}_m \left[ \sum_{k=1}^{K} \mathbf{A}_{mqk} \cdot \mathbf{W}'_m \mathbf{f}\left(\mathbf{p}_q + \Delta \mathbf{p}_{mqk}\right) \right], \tag{3}$$

where $k$ is an index of sampled key vectors, and $K$ is the total number of the key. The attention weight $\mathbf{A}_{mqk}$ of $m$-th attention head has a value between 0 and 1, normalized by $\sum_{k=1}^{K} \mathbf{A}_{mqk} = 1$ similar to the multi-head attention. $\Delta \mathbf{p}_{mqk} = \mathbf{W}_{mk} \mathbf{z}_q$ is a sampling offset of the $k$-th sampling key of the $m$-th attention head. Since the deformable attention module uses only a small set of informative keys for each query, it has linear complexity in the number of inputs.

In recent, Transformer-based models Ying et al. (2021); Dwivedi & Bresson (2020) have been proposed to harness the power of the standard multi-head attention for representing nodes in graphs. Although they have shown superior performance on various tasks, they have difficulty in learning representations on large-scale graphs. Since the large-scale graph has a vast number of nodes, the scalability issue of the multi-head attention is dramatically exacerbated. Also, the enormous number of keys to attend per query node increases the risk of the aggregation of noisy information from irrelevant key nodes. In this paper, we design a sparse attention module, Deformable Graph Attention, that attends to only a small number of relevant keys different from standard attention that blindly aggregates irrelevant key nodes. By reducing the number of key nodes, it is possible to make transformer-based architecture effective and efficient on large-scale graphs.

## 3.2 DEFORMABLE GRAPH ATTENTION

In this section, we present Deformable Graph Attention (DGA), a key module in our proposed Deformable Graph Transformer (DGT). The overview of the deformable graph attention is illustrated in Figure 1. The general idea of deformable graph attention is to design the deformable attention

mechanism for graph-structured data, and therefore DGT performs the attention mechanism with partially selected relevant key nodes, which makes the transformer-based architecture efficient and effective on large-scale graphs. However, extending the deformable attention mechanism to graph representation learning is non-trivial since the offset-based interpolation in the deformable attention only works in Euclidean space whereas graphs are non-Euclidean data. Also, even when embedding nodes in a graph into a low-dimensional space via graph embedding methods (e.g., Node2vec (Grover & Leskovec, 2016)), nodes are irregularly distributed in the low-dimensional space, which makes difficult to effectively deform an attention map.

To address this challenge, we propose a NodeSort module that converts a graph into a sorted sequence of nodes in a regular space. We define a *base node* $v_b$ as the first node in a sorted sequence which is similar to an ego node in an ego graph. NodeSort differentially sorts nodes depending on the base node. In other words, NodeSort provides a relative ordering that varies across base nodes whereas conventional topological sort yields a single (absolute) ordering for a graph. Specifically, given a *base node*, NodeSort sorts nodes and returns a sequence of their features as follows:

$$S_{\pi b} = \text{NodeSort}_\pi(\mathcal{G}, v_b, \{\mathbf{x}_i\}_{i=1}^N) = [\mathbf{x}_{\sigma_{\pi b}^{-1}(i)}]_{i=1}^N, \tag{4}$$

where $\pi$ denotes a specific criterion for sorting nodes and $\sigma_{\pi b}$ is a bijective mapping from $\mathcal{V}$ to $\mathcal{V}$ depending on a base node $v_b$. To consider both structural and semantic proximity, we generate multiple sorted node sequences for each node $v_b$, $\{S_{\pi b}\}_{\pi \in \Pi}$, from a set of diverse criteria $\Pi$, such as Personalized PageRank (PPR) score, BFS, and feature similarity. More details on the NodeSort module and criteria are in Section B.2. Note that $\sigma_{\pi b}$ redefines neighbors of each node $v_b$ based on various aspects $\pi \in \Pi$ beyond 1-hop neighbors in the existing GNNs.

Now, we introduce our Deformable Graph Attention (DGA), which is a sparse attention by dynamically sampling key/value pairs from the set of sorted sequences of node features. To benefit from various properties of the graphs, the deformable graph attention module is designed to deal with diverse node sequences. Different from existing methods based on deformable sampling modules in computer vision, which only considers spatial proximity on a grid, DGA captures both structural and semantic proximity. Given the set of sorted sequences $\{S_{\pi q}\}_{\pi \in \Pi}$ and features of query node $\mathbf{z}_q$, DGA is defined as

$$\text{DGA}(\mathbf{z}_q, \{S_{\pi q}\}_{\pi \in \Pi}) = \sum_{\pi \in \Pi} \sum_{m=1}^M \mathbf{W}_{\pi m} \left[ \sum_{k=1}^K \mathbf{A}_{\pi m q k} \cdot \mathbf{W}'_{\pi m} \tilde{S}_{\pi q}(\Delta \mathbf{p}_{\pi m q k}) \right], \tag{5}$$

where $\tilde{S}_{\pi q}(\Delta \mathbf{p}_{\pi m q k})$ denotes the representation of the $k$-th key, $K$ denotes the number of keys, $\mathbf{W}_{\pi m} \in \mathbb{R}^{c \times c_v}$ and $\mathbf{W}'_{\pi m} \in \mathbb{R}^{c_v \times c}$ are the learnable weight matrices for each criteria $\pi \in \Pi$ and the $m$-th attention head, $\mathbf{A}_{\pi m q k} = \theta^{\text{att}}_{\pi m k}(\mathbf{z}_q)$ denotes the attention weight from a linear function $\theta^{\text{att}}$ between the $q$-th query and the $k$-th key of the $m$-th attention head and criterion $\pi$, which is normalized by $\sum_{k=1}^K \mathbf{A}_{\pi m q k} = 1$ and the attention weight needs to be in a interval $[0, 1]$.

$\Delta \mathbf{p}_{\pi m q k}$ is a sampling offset of the $k$-th key for criterion $\pi$ and $m$-th attention head, which is generated by $\theta^{\text{off}}_{\pi m k}(\mathbf{z}_q)$, where $\theta^{\text{off}}$ is a linear function with an activation function. As the offset $\Delta \mathbf{p}_{\pi m q k}$ is fractional, we compute $\tilde{S}_{\pi q}(\mathbf{p})$ by kernel-based interpolation:

$$\tilde{S}_{\pi q}(\mathbf{p}) = \sum_i g(\mathbf{p}, i) \cdot S_{\pi q}[i], \quad g(a, b) = \begin{cases} \exp\left(-\frac{(a-b)^2}{\gamma}\right), & \text{if } |a - b| < \epsilon \\ 0, & \text{otherwise} \end{cases} \tag{6}$$

where $\gamma \in \mathbb{R}^{++}$ denotes the bandwidth of the kernel, $\epsilon$ is a hyper-parameter for truncating the kernel, and $S_{\pi q}[i]$ is the node feature at a $i$-th index of the sequence $S_{\pi q}$.

A standard way of computing $\tilde{S}_{\pi q}(\mathbf{p})$ is calculating Eq. (6) by substituting $g$ with $g(a, b) = \max(0, 1 - |a - b|)$. But, the existing way considers only two nodes whose coordinates are ceiling and floor values of a point $\mathbf{p}$. It might not be sufficient for the offsets $\Delta \mathbf{p}$ to move to get relevant information by looking at only two nodes. As the scale of the graph increases, a target node requires much more nodes to learn its representation. So, we apply the Radial Basis Function kernel for computing $\tilde{S}_{\pi q}$ as in Eq. (6) to look at a wide range of nodes for representing query nodes.

### 3.3 KATZ POSITIONAL ENCODING

Positional Encoding is a crucial component in Transformer to reflect domain-specific positional information into its attention mechanism. A major issue with positional encoding on graphs is the absence of absolute positions of nodes, unlike other domains. One remedy to this issue is to encode relations between nodes. Here, we propose learnable positional encodings, Katz PE, for Transformer-based graph models based on connectivity between nodes. To be specific, inspired by the matrix of Katz indices (Katz, 1953) which counts all paths between nodes with the decaying weight $\beta$ to reflect the preference for shorter paths, i.e., $\hat{\mathbf{A}} = \sum_{k=1}^{\infty} \beta^{k-1} \mathbf{A}^k$, our method learns positional embeddings, $\text{PE}_i$, of each node $i$ by the nonlinear transform of $\hat{\mathbf{A}}$ as follows:

$$\text{Katz PE}(v_i) = \text{MLP}(\hat{\mathbf{A}}[v_i]^T), \tag{7}$$

where MLP is a Multi-Layer Perceptron, and $\hat{\mathbf{A}}[v_i]$ is the row vector of node $v_i$ in $\hat{\mathbf{A}}$. We limit the maximum $k$ in $\hat{\mathbf{A}}$ to $K$, i.e., $\hat{\mathbf{A}} = \sum_{k=1}^{K} \beta^{k-1} \mathbf{A}^k$ for the efficient calculation. In addition, when $N$ is large, then we sample $N'$ anchor nodes with a high degree and utilize the submatrix of Katz indices $\hat{\mathbf{A}}' \in \mathbb{R}^{N \times N'}$. Our learnable Katz PE is simple yet more effective for both our DGT and vanilla Transformer than existing pre-computed positional encodings. See Section 4.3, for more details.

### 3.4 DEFORMABLE GRAPH TRANSFORMER

Finally, we introduce our Deformable Graph Transformer (DGT) built upon our proposed Graph Deformable Attention and Positional Encoding. Deformable Graph Transformer first encodes node feature $\mathbf{x}_i$ with the learnable function $f_\theta$, which can be MLP, and combines with positional embeddings from Eq. (7) as

$$\mathbf{z}_i^{(0)} = f_\theta(\mathbf{x}_i) + \text{Katz PE}(v_i). \tag{8}$$

Then, given a set of sorted sequences $\{S_q^\pi\}_{\pi \in \Pi}$ from Eq. (5), each $l$-th Deformable Graph Attention layer in DGT performs the attention mechanism with a small set of informative keys and applies skip-connection and MLP to update node representations as follows:

$$\hat{\mathbf{z}}_i^{(l)} = \text{DGA}\left(\mathbf{z}_i^{(l)}, \{S_{\pi i}\}_{\pi \in \Pi}, \mathbf{Z}^{(l-1)}\right) + \mathbf{z}_i^{(l-1)}, \quad \mathbf{z}_i^{(l)} = \text{MLP}(\hat{\mathbf{z}}_i^{(l)}) + \hat{\mathbf{z}}_i^{(l)}. \tag{9}$$

After the stack of $L$ Deformable Graph Attention blocks, each node representation $\mathbf{z}_i^{(L)}$ is used for node classification on top and MLP followed by a softmax layer is used as $\hat{y}_i = \text{Softmax}(\text{MLP}(\mathbf{z}_i^{(L)}))$. Our loss function is a cross-entropy on nodes that have ground truth labels.

### 3.5 COMPLEXITY ANALYSIS

We provide the comparison of computational complexity between the self-attention in most Transformer-based graph models and the Deformable Graph Attention (DGA) in our DGT. As the number of nodes $N$ increases, our DGA with a *linear* complexity of $N$ is more efficient than the self-attention with a *quadratic* complexity of $N$. The details are in Sec A.

## 4 EXPERIMENTS

In this section, we evaluate the effectiveness of our proposed Deformable Graph Transformer (DGT) against state-of-the-art models on node classification benchmark datasets.

### 4.1 EXPERIMENTAL SETUP

**Datasets.** We validate the effectiveness of our model on node classification using four heterophilic graph datasets and four homophilic graph datasets, which are distinguished by the edge-based homophily ratio (Zhu et al., 2020) defined as $h = \frac{|\{(v_i,v_j):(v_i,v_j)\in\mathcal{E} \land y_i=y_j\}|}{|\mathcal{E}|}$. Each dataset has an edge-based homophily ratio ranged from $h = 0.22$ (very heterophilic) to $h = 0.81$ (very homophilic). For large-scale graphs, we evaluate our method on twitch-gamers, obgn-arxiv, and Reddit datasets. More details about the datasets are in Sec F.1.

Table 1: Evaluation results on node classification task (Mean accuracy (%) $\pm$ 95% confidence interval). OOM denotes 'out-of-memory'. **Bold** indicates the model with the best performance and underline indicates the second best model.

| Model | Actor | Squirrel | Chameleon | Cora | Citeseer | twitch-gamers | ogbn-arxiv | Reddit | Avg. Rank |
|---|---|---|---|---|---|---|---|---|---|
| # Nodes | 7,600 | 5,201 | 2,277 | 2,708 | 3,327 | 168,114 | 169,343 | 232,965 | |
| # Edges | 26,659 | 198,353 | 31,371 | 5,278 | 4,552 | 6,797,557 | 1,166,243 | 11,606,919 | |
| Hom. ratio $h$ | 0.22 | 0.22 | 0.23 | 0.81 | 0.74 | 0.55 | 0.66 | 0.76 | |
| *GNN-based Models* | | | | | | | | | |
| MLP | $35.05_{\pm0.38}$ | $31.66_{\pm0.82}$ | $47.11_{\pm0.71}$ | $75.10_{\pm0.84}$ | $73.54_{\pm0.69}$ | $61.14_{\pm0.06}$ | $53.89_{\pm0.21}$ | $70.03_{\pm0.16}$ | 12.38 |
| GCN | $30.13_{\pm0.30}$ | $50.42_{\pm0.66}$ | $66.33_{\pm0.64}$ | $87.22_{\pm0.26}$ | $76.08_{\pm0.43}$ | $64.34_{\pm0.12}$ | $71.27_{\pm0.11}$ | $95.06_{\pm0.03}$ | 7.25 |
| GAT | $30.25_{\pm0.39}$ | $54.26_{\pm1.21}$ | $66.85_{\pm0.88}$ | $86.21_{\pm0.43}$ | $75.71_{\pm0.42}$ | $62.90_{\pm0.22}$ | $70.92_{\pm0.11}$ | OOM | 9.13 |
| GraphSAGE | $35.24_{\pm0.49}$ | $43.75_{\pm0.75}$ | $63.28_{\pm0.68}$ | $86.94_{\pm0.36}$ | $76.25_{\pm0.53}$ | $64.73_{\pm0.11}$ | $70.19_{\pm0.11}$ | $\underline{96.27_{\pm0.01}}$ | 7.13 |
| JKNet | $30.39_{\pm0.35}$ | $55.17_{\pm0.62}$ | $67.81_{\pm0.86}$ | $87.17_{\pm0.33}$ | $76.33_{\pm0.53}$ | $65.08_{\pm0.07}$ | $71.00_{\pm0.15}$ | $95.28_{\pm0.02}$ | 6.25 |
| SGC | $29.43_{\pm0.41}$ | $35.07_{\pm0.51}$ | $49.95_{\pm1.15}$ | $87.33_{\pm0.39}$ | $75.47_{\pm0.56}$ | $60.47_{\pm0.14}$ | $66.56_{\pm0.01}$ | $94.72_{\pm0.00}$ | 11.25 |
| GATv2 | $30.54_{\pm0.41}$ | $57.41_{\pm0.94}$ | $67.25_{\pm0.58}$ | $86.10_{\pm0.41}$ | $75.63_{\pm0.49}$ | $64.15_{\pm0.09}$ | $71.01_{\pm0.15}$ | OOM | 8.25 |
| MixHop | $35.79_{\pm0.33}$ | $38.78_{\pm0.86}$ | $59.27_{\pm0.83}$ | $87.16_{\pm0.38}$ | $75.95_{\pm0.57}$ | $65.20_{\pm0.12}$ | $\underline{71.47_{\pm0.15}}$ | $96.23_{\pm0.04}$ | 6.25 |
| Geom-GCN | $31.53_{\pm0.31}$ | $37.98_{\pm0.42}$ | $60.70_{\pm0.91}$ | $85.38_{\pm0.55}$ | $76.57_{\pm0.56}$ | N/A | N/A | N/A | 10.50 |
| H2GCN | $35.32_{\pm0.34}$ | $36.89_{\pm0.80}$ | $58.21_{\pm0.70}$ | $\mathbf{87.73_{\pm0.64}}$ | $\underline{76.88_{\pm0.54}}$ | OOM | OOM | OOM | 8.50 |
| DeformableGCN | $36.53_{\pm0.42}$ | $62.09_{\pm0.68}$ | $71.03_{\pm0.57}$ | $87.32_{\pm0.44}$ | $76.67_{\pm0.43}$ | OOM | $70.22_{\pm0.19}$ | OOM | 6.00 |
| *Transformer-based Graph Models* | | | | | | | | | |
| Transformer | $36.61_{\pm0.39}$ | $31.00_{\pm0.60}$ | $45.93_{\pm0.83}$ | $73.75_{\pm0.71}$ | $72.99_{\pm0.61}$ | OOM | OOM | OOM | 12.63 |
| Graphormer | $36.54_{\pm0.44}$ | $36.25_{\pm0.72}$ | $50.15_{\pm1.26}$ | $73.44_{\pm0.90}$ | $72.60_{\pm0.63}$ | OOM | OOM | OOM | 12.00 |
| GT-full | $34.53_{\pm0.38}$ | $32.33_{\pm0.64}$ | $49.07_{\pm1.25}$ | $69.51_{\pm1.01}$ | $70.18_{\pm0.67}$ | OOM | OOM | OOM | 13.63 |
| GT-sparse | $34.69_{\pm0.35}$ | $44.22_{\pm0.67}$ | $64.82_{\pm0.57}$ | $85.63_{\pm0.44}$ | $75.49_{\pm0.58}$ | $63.09_{\pm0.71}$ | $71.45_{\pm0.14}$ | OOM | 8.75 |
| **DGT-light (Ours)** | $\underline{36.86_{\pm0.53}}$ | $62.58_{\pm0.57}$ | $73.04_{\pm0.65}$ | $86.60_{\pm0.60}$ | $75.72_{\pm0.40}$ | $\underline{65.59_{\pm0.25}}$ | $71.18_{\pm0.13}$ | $96.14_{\pm0.05}$ | 4.38 |
| **DGT (Ours)** | $\mathbf{36.93_{\pm0.39}}$ | $\mathbf{63.78_{\pm0.59}}$ | $\mathbf{73.48_{\pm0.88}}$ | $\underline{87.55_{\pm0.59}}$ | $\mathbf{77.04_{\pm0.57}}$ | $\mathbf{66.09_{\pm0.22}}$ | $\mathbf{71.77_{\pm0.10}}$ | $\mathbf{96.32_{\pm0.02}}$ | **1.13** |

Table 2: Efficiency comparisons on Transformer-based graph models. $\dagger$ denotes the performance measured by CPU implementation.

| | Chameleon | | Cora | | Citeseer | | Squirrel | | twitch-gamers | | ogbn-arxiv | |
|---|---|---|---|---|---|---|---|---|---|---|---|---|
| # Nodes | 2,277 | | 2,708 | | 3,327 | | 5,201 | | 168,114 | | 169,343 | |
| # Edges | 31,371 | | 5,278 | | 4,552 | | 198,353 | | 6,797,557 | | 1,166,243 | |
| Model | FLOPs | Acc. | FLOPs | Acc. | FLOPs | Acc. | FLOPs | Acc. | FLOPs | Acc. | FLOPs | Acc. |
| Transformer | 1.06G | $45.93_{\pm0.83}$ | 1.26G | $73.75_{\pm0.71}$ | 2.29G | $72.99_{\pm0.61}$ | 4.29G | $31.00_{\pm0.60}$ | $3622G^\dagger$ | $59.85^\dagger$ | OOM | OOM |
| Graphormer | 1.78G | $50.15_{\pm1.26}$ | 2.26G | $73.44_{\pm0.90}$ | 3.79G | $72.60_{\pm0.63}$ | 7.88G | $36.25_{\pm0.72}$ | OOM | OOM | OOM | OOM |
| GT-full | 1.07G | $49.07_{\pm1.25}$ | 1.27G | $69.51_{\pm1.01}$ | 2.31G | $70.18_{\pm0.67}$ | 4.31G | $32.33_{\pm0.64}$ | $3623G^\dagger$ | $59.18^\dagger$ | OOM | OOM |
| GT-sparse | **0.43G** | $64.82_{\pm0.57}$ | 0.43G | $85.63_{\pm0.44}$ | 0.99G | $75.49_{\pm0.58}$ | 1.49G | $44.22_{\pm0.67}$ | 17.04G | $63.09_{\pm0.71}$ | 20.24G | $71.45_{\pm0.14}$ |
| **DGT-light (Ours)** | **0.43G** | $73.04_{\pm0.65}$ | **0.36G** | $86.60_{\pm0.60}$ | **0.87G** | $75.72_{\pm0.40}$ | **1.24G** | $62.58_{\pm0.57}$ | **8.05G** | $65.59_{\pm0.25}$ | **5.02G** | $71.18_{\pm0.13}$ |
| **DGT (Ours)** | 0.49G | $\mathbf{73.48_{\pm0.88}}$ | 0.65G | $\mathbf{87.55_{\pm0.59}}$ | 1.05G | $\mathbf{77.04_{\pm0.57}}$ | 2.63G | $\mathbf{63.78_{\pm0.59}}$ | 16.19G | $\mathbf{66.09_{\pm0.22}}$ | 6.66G | $\mathbf{71.77_{\pm0.10}}$ |

**Baselines.** To demonstrate the effectiveness of our Deformable Graph Transformer (DGT), we compare DGT with following baselines: (1) six standard GNNs: GCN (Kipf & Welling, 2017), GAT (Veličković et al., 2018), GraphSAGE (Hamilton et al., 2017), JKNet (Xu et al., 2018), SGC (Wu et al., 2019), GATv2 (Brody et al., 2022); (2) four GNNs designed for heterophilic settings: MixHop (Abu-El-Haija et al., 2019), Geom-GCN (Pei et al., 2020), H2GCN (Zhu et al., 2020), DeformableGCN (Park et al., 2022); (3) four Transformer-based architectures for graphs: Transformer (Vaswani et al., 2017), Graphormer (Ying et al., 2021), GT-full (Dwivedi & Bresson, 2020), GT-sparse (Dwivedi & Bresson, 2020). (4) two variants of our proposed DGT: DGT-light using a single sorting criterion ($\Pi : $ BFS) and DGT using multiple sorting criteria ($|\Pi| = 3$).

## 4.2 EXPERIMENTAL RESULTS

Table 1 summarizes node classification results of our proposed DGT and DGT-light, GNN-based models, and Transformer-based graph models. DGT consistently shows superior performance across all eight datasets. Especially, our DGT significantly outperforms transformer-based baselines by a large margin up to 106%. Surprisingly, existing Transformer-based graph models show poor performance compared to GNN baselines except for Actor. This implies that Transformer-based graph models failed to filter out irrelevant messages and focus on useful nodes. In addition, they are not

Table 3: Performance comparisons on different ordering and criteria $\pi$ for constructing node sequences.

| Name | Ordering | Criteria ($\pi$) | Cora | Citeseer | Chameleon | Squirrel |
|------|----------|------------------|------|----------|-----------|----------|
| AR | Absolute | Random | $81.53_{\pm1.21}$ | $72.27_{\pm0.72}$ | $70.51_{\pm0.45}$ | $62.44_{\pm0.62}$ |
| AB | Absolute | BFS | $81.22_{\pm1.14}$ | $72.28_{\pm0.69}$ | $71.95_{\pm0.49}$ | $62.31_{\pm0.89}$ |
| AM | Absolute | BFS,PPR,Feat | $83.28_{\pm0.77}$ | $70.20_{\pm0.84}$ | $72.41_{\pm0.57}$ | $62.28_{\pm0.81}$ |
| RB | Relative | BFS | $86.60_{\pm0.60}$ | $75.72_{\pm0.40}$ | $73.04_{\pm0.65}$ | $62.58_{\pm0.57}$ |
| RM | Relative | BFS,PPR,Feat | $\mathbf{87.55_{\pm0.59}}$ | $\mathbf{77.04_{\pm0.57}}$ | $\mathbf{73.48_{\pm0.88}}$ | $\mathbf{63.78_{\pm0.59}}$ |

Table 4: Effects of different positional encodings.

| Model | Positional Encoding | Cora | Citeseer | Chameleon | Squirrel |
|-------|---------------------|------|----------|-----------|----------|
| Transformer | w/o PE | $73.75_{\pm0.71}$ | $72.99_{\pm0.61}$ | $45.93_{\pm0.83}$ | $31.00_{\pm0.60}$ |
| | Node2Vec | $81.52_{\pm0.68}$ | $72.07_{\pm0.58}$ | $49.00_{\pm0.82}$ | $39.15_{\pm0.53}$ |
| | Laplacian Eigvecs | $69.51_{\pm1.01}$ | $70.18_{\pm0.67}$ | $49.07_{\pm1.25}$ | $32.33_{\pm0.64}$ |
| | RWPE | $79.46_{\pm0.55}$ | $71.94_{\pm0.63}$ | $53.81_{\pm0.65}$ | $40.82_{\pm0.68}$ |
| | **Katz PE (ours)** | $81.44_{\pm1.16}$ | $73.02_{\pm0.71}$ | $72.13_{\pm0.60}$ | $59.79_{\pm0.76}$ |
| DGT (ours) | w/o PE | $86.32_{\pm0.52}$ | $76.61_{\pm0.61}$ | $59.44_{\pm0.87}$ | $46.59_{\pm0.75}$ |
| | Node2Vec | $86.80_{\pm0.51}$ | $75.36_{\pm0.54}$ | $62.02_{\pm0.63}$ | $50.26_{\pm0.54}$ |
| | Laplacian Eigvecs | $83.94_{\pm0.67}$ | $76.21_{\pm0.61}$ | $58.02_{\pm0.91}$ | $43.86_{\pm0.67}$ |
| | RWPE | $87.08_{\pm0.60}$ | $76.71_{\pm0.56}$ | $61.48_{\pm0.83}$ | $49.56_{\pm0.67}$ |
| | **Katz PE (ours)** | $\mathbf{87.55_{\pm0.59}}$ | $\mathbf{77.04_{\pm0.57}}$ | $\mathbf{73.48_{\pm0.88}}$ | $\mathbf{63.78_{\pm0.59}}$ |

applicable to large-scale graphs such as ogbn-arxiv, twitch-gamers, and Reddit due to their huge computational costs.

On the other hand, our DGT consistently outperforms both GNNs and Transformer-based graph models on almost all datasets and efficiently handles large-scale graphs by utilizing a small set of relevant nodes. GNNs generally perform well in homophilic graphs such as Cora and Citeseer, but show relatively inferior performance in heterophilic graphs such as Actor, Squirrel, Chameleon, and twitch-gamers. This is because most GNNs utilize directly connected nodes for aggregation even in heterophilic graphs. Instead, our DGT and DGT-light show larger performance gain in heterophilic graph datasets since it captures long-range dependency, which is important for learning on heterophilic graphs.

We also compare our DGT and DGT-light with other Transformer-based architectures to validate the efficiency (FLOPs) of our DGT in Table 2. The table shows that our DGT and DGT-light improves not only the performance but also the efficiency with deformable graph attention on various datasets. As the number of nodes increases, the gap between Transformer and variants of DGT with respect to FLOPS becomes bigger. In particular, on the twitch-gamers, DGT-light (8.05G) reduces computational costs by $\times$ 449 compared to Transformer (3622G).

### 4.3 QUANTITATIVE ANALYSIS

Here, we provide quantitative results of additional experiments to demonstrate the contribution of each component of our Deformable Graph Transformer. We first provide the ablation study of the NodeSort module and examine the effectiveness of our positional encoding comparing with popular positional encoding methods.

**NodeSort module.** We conduct an ablation study of the NodeSort module to study the effect of the *relative* ordering that varies depending on *base nodes* and various sorting criteria. We compare several constructions: absolute ordering with random permutation (AR), absolute ordering with BFS (AB), absolute ordering with multiple criteria (AM), relative ordering with BFS (RB), and relative ordering with multiple criteria (RM). From Table 3, the absolute ordering approaches (AR, AB, AM) show poor performance. The absolute ordering with BFS (AB) shows no performance gain compared to a randomly permuted absolute ordering (AR) on three datasets. This shows that a

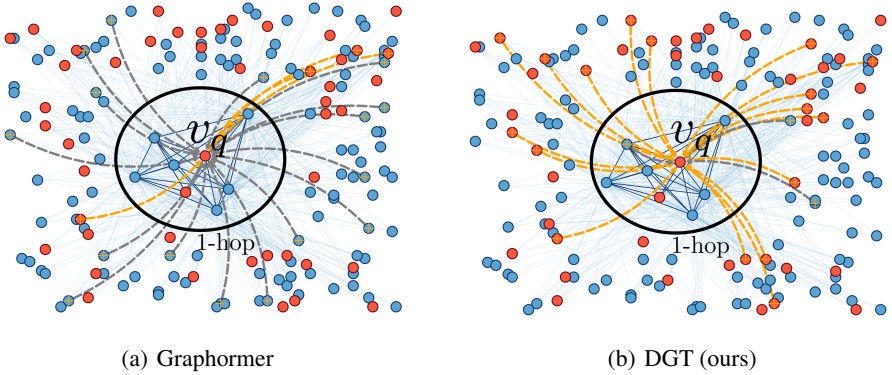

(a) Graphormer             (b) DGT (ours)

Figure 2: Visualization of the 20 most important key nodes for a given query node $v_q$ in (a) Graphormer and (b) DGT on Chameleon validation set. Red nodes denote nodes with the same label as the query node's label whereas blue nodes have different labels. Orange dashed lines represent connections between the query node and key nodes with the same label as the query node and gray dashed lines represent the connections between nodes with different labels.

single absolute node sequence is not sufficient to learn complex relationships between nodes. On the other hand, the relative ordering with BFS (RB) shows a significant performance improvement up to 5.07(%). Further, the relative ordering with multiple criteria (RM) consistently shows superior performance compared to the relative ordering with BFS (RB).

**Positional encoding.** To validate the effectiveness of our positional encoding, we compare our proposed PE methods with models without positional encodings (w/o PE) and various encoding methods such as Node2Vec (Grover & Leskovec, 2016), Laplacian Eigvecs used in GT (Dwivedi & Bresson, 2020), and RWPE Dwivedi et al. (2022) on four datasets. We use Transformer and DGT for the base models. Table 4 demonstrates that our positional encoding is effective on both base models. In particular, it improves the performance by 17.19(%) compared to DGT without positional encoding on Squirrel.

## 4.4 QUALITATIVE ANALYSIS

We provide qualitative analysis to understand why DGT is effective. We visualize the top 20 key nodes with high attention scores for a given query node $v_q$ in Graphormer and DGT. In DGT, we compute attention scores of each node, $v_i$ by $w_i = \sum_k \mathbf{A}_{\pi mqk} \cdot g(\Delta \mathbf{p}_{\pi mqk}, i)$. As shown in Figure 2, a within 1-hop neighborhood of the given query node $v_q$, 6 out of 7 neighbors have different labels. So, both DGT and Graphormer aggregate messages (dashed lines) from remote nodes beyond 1-hop neighbors through the attention mechanism. 17 of the top 20 nodes in DGT are nodes with the same label whereas only 4 out of 20 nodes have the same label in Graphormer. Also, the ratio of the attention scores for the nodes with the same label, *i.e.*, $\sum_{\{v_i:v_i \in \mathcal{V} \wedge y_q = y_i\}} w_i / \sum_{v_i \in \mathcal{V}} w_i$ is 0.97 for DGT and 0.21 for Graphormer. This evidences that our DGT effectively performs the sparse attention by focusing on a small set of relevant key nodes compared to Graphormer.

## 5 CONCLUSION

We propose Deformable Graph Transformer (DGT) that performs sparse attention, named Deformable Graph Attention (DGA) for learning node representations on large-scale graphs. With Deformable Graph Attention, our model can address two limitations of Transformer-based graph models such as a scalability issue and aggregation of noisy information. Different from standard deformable attention (Zhu et al., 2021; Xia et al., 2022), the Deformable Graph Attention considers both structural and semantic proximity based on diverse node sequences. Also, we design simple and effective positional encodings for graph Transformers. Our extensive experiments demonstrate that DGT outperforms existing Transformer-based graph models on eight graph datasets. We hope our work paves the way for generalizing Transformers on large-scale graphs.

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

## A    COMPLEXITY FOR DEFORMABLE GRAPH ATTENTION

**Self-Attention in most Transformer-based graph models (Vaswani et al., 2017; Ying et al., 2021; Dwivedi & Bresson, 2020).** Suppose that $N$ is the number of nodes, $C$ is the dimensionality of hidden representations. The self-attention operation requires a huge computation cost with the complexity of $\mathcal{O}\left(N^2C + NC^2\right)$, where $C$ is the dimensionality of hidden representations.

**Deformable Graph Attention.** Consider $M$ is the number of heads, $K$ is the number of keys, $T$ is the number of critera, and $W$ is the number of nonzero values of $g$ in Eq. (6). In the Deformable Graph Attention (Eq. (5)), calculating the sampling offsets $\Delta\mathbf{p}_{\pi mqk}$ and attention weights $\mathbf{A}_{\pi mqk}$ requires the complexity of $\mathcal{O}(NCMKT)$. Then, given the sampling offsets and attention weights, the complexity of calculating Eq. (5) is $\mathcal{O}(NC^2T + NKC^2T + WNKCT)$, where $W$ is the number of nonzero values of $\tilde{S}_{\pi q}(\Delta\mathbf{p}_{\pi mqk})$, and the factor $W$ is because of the kernel-based interpolation. We can calculate the linear transformation of the interpolated features $\mathbf{W}'_{\pi m}\tilde{S}_{\pi q}(\Delta\mathbf{p}_{\pi mqk})$ by interpolation of the linear transformed features $\mathbf{W}'_{\pi m}\mathbf{X}$. So, the complexity of calculating Eq. (5) become as $\mathcal{O}(NC^2T + WNKCT)$. In the results, the overall complexity of Deformable Graph Attention is $\mathcal{O}(N \cdot (C^2T + WKCT + CMKT))$. In our implementation, we set $M = 4, K = 4$ and $C = 64$ as a default, thus $MK < C$ and the complexity is of $\mathcal{O}(N \cdot (C^2T + WKCT))$ ,which is a linear complexity with respect to the number of nodes $N \gg W, K, C, T$.

## B    ADDITIONAL EXPERIMENTS

Here, we provide additional experimental results to analyze the contributions of each component of our Deformable Graph Transformer (DGT) including the ablation study of the NodeSort module and our kernel-based interpolation.

### B.1    NODESORT MODULE.

We conduct an ablation study for the NodeSort module to validate the effectiveness of a single node sequence with each criterion and multiple node sequences. We use three criteria for sorting: Breadth-first Search (BFS), Personalized PageRank score (PPR), and Feature similarity (Feature). As shown in Table 5, when multiple node sequences are applied, DGT shows good performance on four datasets. In particular, on the cora dataset, multiple node sequences improve 3.46 (%) over the model with the Feature similarity criterion.

Table 5: Performance comparisons on different criteria for constructing node sequences.

| Criteria | | | Squirrel | Chameleon | Cora | Citeseer |
|---|---|---|---|---|---|---|
| BFS | PPR | Feature | | | | |
| ✓ | | | $62.58_{\pm0.57}$ | $73.04_{\pm0.65}$ | $86.60_{\pm0.60}$ | $75.72_{\pm0.40}$ |
| | ✓ | | $62.57_{\pm0.59}$ | $72.66_{\pm0.77}$ | $86.31_{\pm0.45}$ | $75.42_{\pm0.64}$ |
| | | ✓ | $62.66_{\pm0.73}$ | $72.67_{\pm0.80}$ | $84.09_{\pm0.62}$ | $75.31_{\pm0.61}$ |
| ✓ | ✓ | ✓ | $\mathbf{63.78_{\pm0.59}}$ | $\mathbf{73.48_{\pm0.88}}$ | $\mathbf{87.55_{\pm0.59}}$ | $\mathbf{77.04_{\pm0.57}}$ |

### B.2    KERNEL-BASED INTERPOLATION.

In Figure 3, we conduct an ablation study of our kernel-based interpolation according to $\epsilon = 1, 2, 3, 4, 5, 6, 7, 8$ in $g$ on the Cora dataset. The model shows the worst performance when $\epsilon = 1$. As the value of $\epsilon$ increases, the model shows better performance. We believe that the value of $\epsilon$ needs to be appropriately big to find out relevant nodes for a query node.

We also compare the performance of bilinear interpolation used in (Zhu et al., 2021) and kernel-based interpolation in Table 6. As shown in the table, our kernel-based interpolation improves learnability of offsets compared to the bilinear interpolation.

Table 6: Performance comparisons on different interpolation methods.

| Method | Cora | Citeseer | Chameleon | Squirrel | twitch-gamers |
|---|---|---|---|---|---|
| Bilinear interpolation | $86.02_{\pm 0.53}$ | $76.61_{\pm 0.56}$ | $68.51_{\pm 0.86}$ | $61.69_{\pm 0.65}$ | $65.30_{\pm 0.15}$ |
| **Kernel-based interpolation (Ours)** | $\mathbf{87.55_{\pm 0.59}}$ | $\mathbf{77.04_{\pm 0.57}}$ | $\mathbf{63.78_{\pm 0.59}}$ | $\mathbf{73.48_{\pm 0.88}}$ | $\mathbf{66.09_{\pm 0.22}}$ |

Table 7: Evaluation results on graph regression task.

| Model | ZINC |
|---|---|
| GCN | 0.367 |
| GAT | 0.384 |
| GraphSAGE | 0.398 |
| GT-full | 0.226 |
| Graphormer | **0.122** |
| DGT (Ours) | 0.158 |

### B.3 Experimental results on graph regression task.

To validate the effectiveness of Deformable Graph Transformer (DGT), we additionally conduct experiment on graph regression task with ZINC dataset Dwivedi et al. (2020) Table 7. Even though DGT mainly focus on node classification tasks, our DGT shows better performance compared to GT-full, which has been proposed for dealing with graph regression/classification tasks.

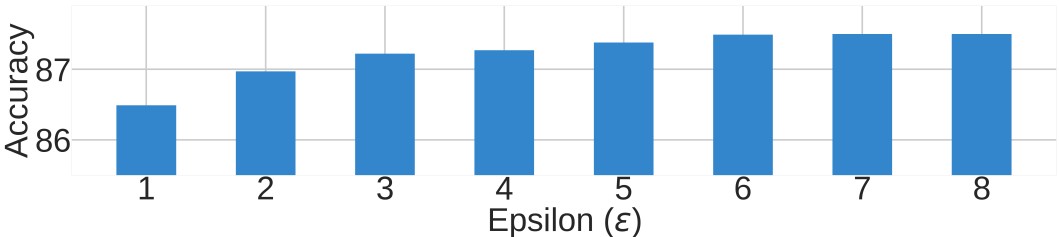

Figure 3: Analysis of our kernel-based interpolation according to $\epsilon$ in $g$.

## C Details of NodeSort Module

Our NodeSort module converts a graph into a sorted sequence of nodes to embed nodes in the regular space without using learnable parameters. The criteria indicate how to sort our nodes. We use three criteria: BFS, Personalized PageRank (PPR) score, and Feature similarity.

- Breadth First Search (BFS): Breadth First Search is a widely used algorithm for searching on graph-structured dataset. Given the graph and the base node, we set the base node as a root node and do breadth first search to get the sequence of nodes.

- Personalized PageRank score : Personalized PageRank score is a score that encodes the local neighborhood for root node. We use approximated personalized pagerank score Klicpera et al. (2019) for NodeSort module. Given the graph and the base node, we calculate personalized page rank score as $\pi_{\text{ppr}}^{(k+1)}(i_x) = (1-\alpha)\hat{A}\pi_{\text{ppr}}^{(k)}(i_x) + \alpha\pi_{\text{ppr}}^{(k)}(i_x)$, where the x is the base node. Then, we sort nodes in descending order of personalized page rank score.

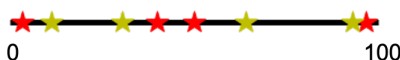

Figure 4: Visualization of the position of $\Delta(\mathbf{p}_{\pi\mathbf{qk}})$ (stars) in Section 3 on sequence generated via BFS with Actor dataset. Color is different according to the query node $v_q$.

- Feature similarity : We use cosine similarity to compute inout feature similarity between nodes. Given the input features of nodes and the base node, we measure the similarity score between the input feature of base node and other nodes through the cosine similarity. Then, we sort nodes in descending order of similarity score.

BFS and PPR are used to generate node sequences based on structural similarity. In BFS, nodes with low shortest path distances from the base node are located in front of the sequence, and in PPR, nodes with high random walk scores from the base node are located in front of the sequence. Conversely, feature similarity is used for semantic proximity between nodes, and nodes with high semantic proximity to the base node are located in front of the sequence. By adopting diverse criteria for NodeSort module, our deformable graph attention simultaneously consider both semantic and structure proximity to design node sequences.

## D    COMPARISON OF KATZPE WITH RWPE

Our Katz PE is different from Random Walk Positional Encoding (RWPE) (Connectivity between two nodes under $k$-th hop v.s. Landing probability of a node to itself on $k$-th hop). We clarify the difference in the below:

Random Walk Positional Encoding (RWPE) of $i$-th node is defined as

$$p_i^{\text{RWPE}} = \left[ \text{RW}_{ii}, \text{RW}_{ii}^2, \cdots, \text{RW}_{ii}^k \right] \in \mathbb{R}^k, \tag{10}$$

where $\text{RW} = AD^{-1}$ is the random walk operator. So, RWPE expresses the landing probability of node $i$ to itself on $k$-th hop. On the other hand, our Katz PEs are defined as

$$\text{KatzPE}(v_i) = \text{MLP}(\hat{A}[v_i]^T), \tag{11}$$

where $\hat{\mathbf{A}} = \sum_{k=1}^K \beta^{k-1} \mathbf{A}^k$ and it represents connectivity between nodes with the decaying weight $\beta$ to reflect the preference of path lengths. The equations clearly show that KatzPE does NOT have $D^{-1}$ and RW does NOT have $\beta^{k-1}$ and summation.

## E    COMPARISON OF DGT WITH DEFORMABLE DETR

Our deformable graph attention addresses the challenges of applying deformable attention to graph-structured data. Since the deformable attention mechanism can only work on the regular space, it is not directly applicable to graph data. We address the challenge by transforming an irregular space into a set of regular spaces using our NodeSort module. Our NodeSort module is novel in the following aspects:

- **Locality** : Deformable attention in Deformable DETR is defined on only one global grid whereas our deformable graph attention utilizes node sequences locally defined on each neighborhood (ego graph).
- **Multiple relations** : Our Deformable Graph Attention (DGA) module uses diverse criteria (e.g. bfs, feature similarity, ppr) even in a neighborhood for constructing multiple node sequences to consider various relations in a graph.
- **Semantic space** : We extended the deformable attention from a physical space to semantic space by performing NodeSort based on node feature similarity.

We also improve learnability of offsets by replacing bilinear interpolation in deformable DETR with our kernel-based interpolation (Eq. (6)). Although this technique is simple, it largely improves the performance as shown in Table 6.

## F  EXPERIMENTAL DETAILS

### F.1  DATASETS

We use four heterophilic graph datasets and four homophilic graph datasets for our experiments. To the best of our knowledge, all the datasets for our experiments **do not contain personally identifiable information or offensive contents**.

For homophilic graphs, we use two Planetoid datasets (Citeseer and Cora) (Sen et al., 2008), OGB node classification dataset (ogbn-arxiv)[1] (Hu et al., 2020), and Reddit (Hamilton et al., 2017). Planetoid and ogbn-arxiv datasets are citation networks whose nodes represent papers and edges indicate citations between papers. Node labels are the topics of each paper and node features are the bag-of-words of papers in Planetoid datasets and word2vec of papers in ogbn-arxiv. Reddit dataset is composed of Reddit posts and the node label is community, which a post belongs to.

For heterophilic graphs, we use four graph datasets: Squirrel[2] (Rozemberczki et al., 2021), Chameleon[2] (Rozemberczki et al., 2021), Actor[3] (Tang et al., 2009), and twitch-gamers[4] (Lim et al., 2021; Rozemberczki & Sarkar, 2021). Squirrel and Chameleon are web page datasets collected from Wikipedia (Rozemberczki et al., 2021), where the nodes are web pages, edges are links between them, node features are keywords of the pages, and labels are five categories based on the monthly traffic of the web pages. Actor is an actor co-occurrence network, where nodes are actors, edges represent co-occurrence on the same Wikipedia page, node features are keywords in the Wikipedia pages, and labels are five categories in terms of words of actor's Wikipedia. twitch-gamers is an online social network (Lim et al., 2021; Rozemberczki & Sarkar, 2021), where nodes are Twitch users, edges are mutual follower relationships between them, and node features include a number of views, creation and update dates, language, life time, and whether the account is dead. Node labels denote whether the channel has explicit content.

### F.2  IMPLEMENTATION DETAILS

As written in Section 4.1 in the main paper, all the models including baselines and our DGT are optimized using Adam optimizer (Kingma & Ba, 2015). The experiments are conducted on a single GPU (RTX 2080 Ti or A6000). For all cases, learning rates and weight decay are optimized in the same search space: learning rate in {0.05, 0.01, 0.005}, weight decay in {1e-3, 5e-4, 5e-5}. The hidden dimension is fixed with 64 and for all the cases. Also, the dropout (Srivastava et al., 2014) is applied and the epochs are 1000 with patience 100 for early stopping. The best model on the validation split is used for reporting the performance. We adopt the splits (48%/ 32%/ 20%) of nodes per class for (train/ validation/ test) following (Pei et al., 2020; Zhu et al., 2020) and the experiments are repeated **30 times** on Actor, Squirrel, Chameleon, Cora, and Citeseer datasets. For twitch-gamers, ogbn-arxiv, and Reddit, the experiments are conducted with the splits provided by (Lim et al., 2021), (Hu et al., 2020), and (Hamilton et al., 2017), respectively and repeated **10 times**.

**Implementation details of Baselines.**  We implement the baselines using Pytorch[5] Paszke et al. (2017), geometric deep learning library Torch-Geometric[6] Fey & Lenssen (2019), and Deep Graph Library[7] Wang et al. (2019). The detailed experimental settings for each baseline models are in Table 8.

---

[1] Copyright (c) 2019 OGB Team. Licensed under MIT License

[2] Copyright (c) 2007 Free Software Foundation, Inc. under GNU GENERAL PUBLIC LICENSE

[3] Copyright (c) Wikipedia:Text of Creative Commons Attribution. under ShareAlike 3.0 Unported License

[4] Copyright (c) 2019 Benedek Rozemberczki. Licensed under MIT License

[5] Copyright (c) 2016- Facebook, Inc (Adam Paszke). Licensed under BSD-3-Clause License

[6] Copyright (c) 2020 Matthias Fey. Licensed under MIT License

[7] Copyright (c) 2019 DGL. Licensed under Apache License, Version 2.0

Table 8: The hyper-parameter settings for our experiments

| Model | Hyper-parameters |
|---|---|
| MLP | learning rate in {0.05, 0.01, 0.005}, weight decay in {1e-3, 5e-4, 5e-5} |
| GCN[8] | learning rate in {0.05, 0.01, 0.005}, weight decay in {1e-3, 5e-4, 5e-5} layer in {1, 2, 3, 4} |
| GAT[9] | learning rate in {0.05, 0.01, 0.005}, weight decay in {1e-3, 5e-4, 5e-5}, layer in {1, 2, 3, 4}, the number of heads in {1,4} |
| GraphSAGE[10] | learning rate in {0.05, 0.01, 0.005}, weight decay in {1e-3, 5e-4, 5e-5}, layer in {1, 2, 3, 4} |
| JKNet | learning rate in {0.05, 0.01, 0.005}, weight decay in {1e-3, 5e-4, 5e-5}, layer in {1, 2, 3, 4} |
| SGC[11] | learning rate in {0.05, 0.01, 0.005}, weight decay in {1e-3, 5e-4, 5e-5}, $K$ in {1, 2, 3, 4} |
| GATv2[12] | learning rate in {0.05, 0.01, 0.005}, weight decay in {1e-3, 5e-4, 5e-5}, layer in {1, 2, 3, 4}, the number of heads in {1,4} |
| MixHop[13] | learning rate in {0.05, 0.01, 0.005}, weight decay in {1e-3, 5e-4, 5e-5}, layer in {1, 2, 3, 4}, and maximum value of $P$ is 2 |
| Geom-GCN[14] | learning rate in {0.05, 0.01, 0.005}, weight decay in {1e-3, 5e-4, 5e-5} |
| H2GCN | learning rate in {0.05, 0.01, 0.005}, weight decay in {1e-3, 5e-4, 5e-5}, $K$ in {1, 2} |
| DeformableGCN | learning rate in {0.05, 0.01, 0.005}, weight decay in {1e-3, 5e-4, 5e-5}, block in {1, 2}, the number of neighbors in {5,10,15,20,25} |
| Transformer[15] | learning rate in {0.05,0.01,0.005}, weight decay in {1e-3,5e-4,5e-5}, the number of blocks in {1,2,3}, the number of heads in {1,4} |
| Graphormer[16] | learning rate in {0.05,0.01,0.005}, weight decay in {1e-3,5e-4,5e-5}, the number of blocks in {1,2,3}, and the number of heads in {1,4} |
| GT-full[17] | learning rate in {0.05,0.01,0.005}, weight decay in {1e-3,5e-4,5e-5}, the number of blocks in {1,2,3}, the number of heads in {1,4} |
| GT-sparse[18] | learning rate in {0.05,0.01,0.005}, weight decay in {1e-3,5e-4,5e-5}, the number of blocks in {1,2,3}, the number of heads in {1,4} |
| **DGT-light (Ours)** | learning rate in {0.05, 0.01, 0.005}, weight decay in {1e-3, 5e-4, 5e-5}, the number of blocks in {1, 2}, $\gamma$ in {1,2,4,8,16,32,64,128,256} |
| **DGT (Ours)** | learning rate in {0.05, 0.01, 0.005}, weight decay in {1e-3, 5e-4, 5e-5}, the number of blocks in {1, 2}, $\gamma$ in {1,2,4,8,16,32,64,128,256} |

**Ours.** We implement our model (DGT and DGT-light) using Pytorch and Torch-Geometric. For constructing node sequences, we employ three criteria such as BFS, sorting based on Personalized PageRank score, and sorting based on feature similarity between a base node and nodes in a graph. The detailed hyperparameter settings are in Table 8.

# G  DISCUSSION

**Negative societal impacts.**  Deformable Graph Transformer (DGT) is a graph Transformer for learning node representations on large-scale graphs. We believe that this paper has no direct negative societal impacts. However, similar to other neural networks for graph-structured datasets such as Graph Neural Networks (GNNs), DGT can be utilized for malicious applications. Graph neural networks can be applied to predict unknowable information such as religions, political views, and personal preferences based on graph information. If this technology is applied to identifying the personalities of voters and influencing their behaviors, it might cause interference in the elections. To mitigate these societal problems, illegal data collection and data harvest should be prevented and benchmark datasets should be released without any private information.

**Limitations.**  Deformable Graph Transformer (DGT) performs deformable attention based on diverse node sequences. The node sequences play the role of coordinates in 2D images. However, different from the works based on deformable attention in computer vision Zhu et al. (2021); Xia et al. (2022), nodes in graphs do not have the exact positions. So, the position of each node needs to be defined with appropriate mechanisms based on the properties of graphs. In this paper, we utilize multiple criteria for generating node sequences to capture various properties of the graphs for node classification. In other tasks such as link prediction and graph classification, other criteria for generating node sequences could lead to more powerful representations on graphs. We leave it for our future work.

