# OpenReview forum: "Deformable Graph Transformer"
_ICLR.cc/2023/Conference — Submitted to ICLR 2023_

### Official Review · Reviewer_KagM · 2022-10-14

**Confidence:** 4
**Correctness:** 3
**Technical Novelty And Significance:** 3
**Empirical Novelty And Significance:** 3
**Recommendation:** 6

**Clarity, Quality, Novelty And Reproducibility:**

Clarity:
- Are the K values the first from the sorting? If so, it could be interesting to space these values since you already interpolate between neighbors.
- If I understand correctly, DGA interpolate on all the nodes of a given sorting for K=4 references. Wouldn't it be worth trading the interpolation for increased K?
- It could be useful to give the size of the models. It seems that the number of parameters in DGT scales linearly with the number of sorting criteria?
- I am not super convinced by the idea proposed in Figure 2: why querying nodes with similar labels necessarily makes an attention mechanism better? Why would only similar nodes be relevant and not dissimilar nodes? In light of this, I think the claim that other methods aggregate information from irrelevant node exaggerated.

Quality:
- The work successfully tackles an important problem and both contributions (DGA and Katz PE) could be reused in subsequent works
- The experiments use several baselines with thorough error bars and both homophilic and heterophilic graphs, with ablations hence are very convincing.

Novelty:
- I have a slight doubt on the novelty of Katz PE. Indeed, it is common in the graph kernel litterature to enumerate paths. Could you comment on this? See e.g. [1].

Reproducibility:
- The code and hyperparameter grid were provided.

[1] Shortest-path kernels on graphs (Borgwardt and Kriegel, 2005).


**Strength And Weaknesses:**

Strengths:
- This work address an important problem, namely how to make graph transformers work when the number of nodes is large.
- The strategy proposed by the authors seems very effective on all datasets both in terms of accuracy and FLOPs.
- The authors provide ablations to study the effectiveness of their two contributions (DGA and Katz PE).

Weakness:
- Although the paper is mostly well-written, I had a hard time understanding how DGA works. In particular, the idea of sampling offsets could be made more clear both in the text and in the figure (I do not understand what do the arrows do in the blue boxes in the center) so that the work is more self-contained.
- The claim that other architectures query irrelevant nodes could be backed more rigorously (see below).
- I am not sure the comparison are fair or at least clear in terms of model size (see below).
- It would be even more convincing to demonstrate that DGT is also good at graph classification/regression tasks

**Summary Of The Paper:**

This work propose DGT, a transformer architecture for dealing with graphs with a large number of nodes.
DGT combines an adaptation of the deformable attention scheme to graphs and a new learnable position encoding for nodes.

The deformable attention mechanisms consists in sorting all the nodes in the graph w.r.t. some criterion viewed from all nodes (e.g., BFS). This is a pre-processing step. Then, for a given query node and each combination of (sorting criterion, head), for each k of the K first nodes, a deformable attention is computed. More precisely, the deformable attention interpolates the values of all the nodes w.r.t. node k in the sorting using the RBF kernel.

On the other hand, the new Katz position encoding counts all paths between a given node and all the others with decaying weight for longer paths. A subsequent processing by a MLP makes it learnable.

Then, the authors demonstrate the superiority of their approach on several homophilic and heterophilic node-classification datasets in terms of accuracy and FLOPs. Some ablations study are performed to demonstrate the usefulness of their relative multi-criteria search and position encoding separately. Finally, the work study the nodes that are selected by their attention mechanism compared to Graphormer.


**Summary Of The Review:**

This work combines two effective mechanisms to propose a graph transformer scaling to large number of nodes with very good performance in terms of accuracy and FLOPs. It also has some defaults such as lack of clarity for one of the two mechanisms (sampling offset in DGA), claims about novelty of Katz PE and about how classical architectures aggregate "irrelevant" nodes that could be better supported. I also think it would be interesting to demonstrate the usefulness of DGT for graph classification.

Overall, I think the pros outweight the cons and I recommend acceptance. I would further raise my score if these concerns are answered.

---

> ### Author Response · Authors · 2022-11-18
> **Response to Reviewer KagM**
>
> We sincerely thank you for your constructive comments for our paper. We address your concerns below.
>
> > Q1. Clarity of the method (DGA). The idea of sampling offsets could be made more clear.
>
> We apologize for the confusion of the word ‘sampling’. We’d like to clarify that we propose a **sparse** attention (Deformable Graph Attention) that can aggregate only some key nodes rather than all key nodes, for each query node. In other words, our model does not sample the input graph in advance but learns to adaptively select only some useful nodes among all nodes. Specifically, we first compute offset $\Delta \mathbf p = \theta_{\pi m k}^{\text{off}}(\mathbf z_q)$ where $\theta^{\text{off}}$ is a linear function with an activation function, for each query node $v_q$. Then, we construct $K$ number of Key representations $\tilde{S}_{\pi q}(\Delta\mathbf{p})$ via kernel-based interpolation with $2\epsilon$ number of nodes as formulated in Eq (5). Finally, we aggregate Keys with the attention module defined in Eq. (4).
>
> > Q2. Wouldn't it be worth trading the interpolation for increased K?
>
> We empirically found that $K=4$ is enough for learning node representations.
>
> > Q3. I am not super convinced by the idea proposed in Figure 2: why querying nodes with similar labels necessarily makes an attention mechanism better? Why would only similar nodes be relevant and not dissimilar nodes? In light of this, I think the claim that other methods aggregate information from irrelevant node exaggerated.
>
>
> Many papers  claimed that graph-based models show good performance on the homophilic graphs where connected nodes have same labels. Since the attention mechanism can be considered as the message passing framework on fully connected graphs, the information of nodes with similar labels is more informative than nodes with dissimilar labels.
>
>
> > Q4. Number of parameters of the model.
>
> We compare our DGT with other Transformer-based architectures to show the size of models in the below tables as you suggested.
>
> |Model| Chameleon | Cora | Citeseer |
> |-----|--|--|--|
> | Transformer |187k | 147k | 292k |
> | Graphormer | 373k | 259k | 549k |
> | GT-full  | 204k | 134k | 285k |
> | GT-sparse | 191k | 130k | 275k |
> | DGT-sparse (ours)  | 206k | 156k | 311k |
> | DGT (ours) | 226k | 176k | 331k |
>
> > Q5. DGT on graph classification/regression tasks
>
> Good suggestion. As you suggested, we additionally conduct experiment on the graph regression task with ZINC dataset in the below table. Even though our DGT mainly focus on node classification tasks, our DGT shows the second best performance compared to the baselines.
>
> |Model| ZINC |
> |-----|--|
> | GCN |0.367 |
> | GAT | 0.384 |
> | SAGE | 0.398 |
> | GT | 0.226 |
> | Graphormer | 0.122 |
> | DGT (ours) |0.158 |
>
> > Q6. Novelty of Katz PE compared to shortest-path kernels on Graph.
>
> Since the shortest-path kernel on graphs  used to measure the similarity between two graphs, it cannot be directly used as a positional encoding. Presumably, the reviewer seems to ask about the difference from the shortest path distance used in the kernel. One of our baselines, Graphormer, used the shortest path distances between nodes as relative positional encoding, and our model with KatzPE showed better performance compared to Graphormer. Also, the computation of the shortest path distance through Floyd-Warshall has a complexity of $O(n^3)$, whereas ours is $O(K \cdot N^2)$  $(K \ll N)$, which shows that KatzPE is more efficient that computation of the shortest path distance.

---

> > ### Comment · Reviewer_KagM · 2022-11-18
> > **Thanks**
> >
> > Thank you for the clarifications. Another important baseline for ZINC would be GPS (Rampasek et al., 2022) which does around $0.07$.
> > On path-based kernel: my point is that other works use paths to provide some structure to the transformer (Graphormer as you mentionned, and e.g. also GraphiT since diffusion reflects the connectivity between two nodes), which should be discussed to better understand why Katz PE is different. As I understand, Katz is similar in spirit to diffusion, because it reflects the connectivity between the nodes, and differs mainly because a MLP is learned on top and the resulting vector is added to the embedding, and not as a term to modulate attention as in GraphiT, do you agree?

---

> > > ### Author Response · Authors · 2022-11-22
> > > **Re: Thanks**
> > >
> > > Thank you for your response. We sincerely appreciate your constructive comments. We respond to your comments below:
> > >
> > > > Another important baseline for ZINC would be GPS (Rampasek et al., 2022) which does around 0.07.
> > >
> > > Thank you for suggestion. We will include GPS as the baseline for ZINC in the final version.
> > >
> > > > On path-based kernel: my point is that other works use paths to provide some structure to the transformer (Graphormer as you mentioned, and e.g. also GraphiT since diffusion reflects the connectivity between two nodes), which should be discussed to better understand why Katz PE is different. As I understand, Katz is similar in spirit to diffusion, because it reflects the connectivity between the nodes, and differs mainly because a MLP is learned on top and the resulting vector is added to the embedding, and not as a term to modulate attention as in GraphiT, do you agree?
> > >
> > > We have provided the comparison of KatzPE with RWPE, which is random-walk-based positional encoding in Section D of our revised version. As suggested, we discuss the difference between Katz PE and GraphIT below.
> > >
> > > To reflect the structural information, GraphIT uses a diffusion kernel, which is defined as follows:
> > >
> > > $$
> > > K_{\text{GraphIT}} = \sum_{i=1}^m e^{-\beta \lambda_i}u_i u_i^\top = e^{-\beta L} = \lim_{p \rightarrow + \infty} \left(I-\frac{\beta}{p} L \right)^p,
> > > $$
> > >
> > > where $L$ is the graph Laplacian matrix. So, the diffusion kernel is a “**graph Laplacian**” based kernel, whereas Katz kernel is an **“adjacency based”** kernel, which is defined as
> > >
> > > $$
> > > \text{Katz PE}(v_i) = \text{MLP}(\hat{\mathbf A} [v_i]^\top), \ \hat{\mathbf{A}}= \sum_{k=1}^K \beta ^{k-1} \mathbf{A}^k.
> > > $$
> > >
> > > In addition, GraphIT uses the diffusion kernel to modulate attention. Specifically, GraphIT uses  the structural similarity of nodes to guide the attention scores as follows:
> > >
> > > $$
> > > \text{Attention}_{\text{GraphIT}}=\text{normalize}\left( \exp( \frac{QQ^\top}{\sqrt d} )\odot K\right),
> > > $$
> > >
> > > where $K$ is a kernel on the graph (***i.e.,*** the diffusion kernel).  On the other hand, we use Katz PE by adding it to the node embedding as below:
> > >
> > > $$
> > > \mathbf z_i^{(0)} = f_\theta (\mathbf x_i) + \text{Katz PE}(v_i).
> > > $$
> > >
> > > Both the Katz kernel and the diffusion kernels are graph kernels that measure the proximity between nodes. So, adopting Katz Kernel in modulating attention similar to GraphIT can be an interesting direction in the future.

---

### Official Review · Reviewer_wsk3 · 2022-10-24

**Confidence:** 4
**Correctness:** 2
**Technical Novelty And Significance:** 2
**Empirical Novelty And Significance:** 1
**Recommendation:** 5

**Clarity, Quality, Novelty And Reproducibility:**

The presentation of method is unclear and hard to follow. More details about this can be found in point 1, 2 of __weaknesses__.
I also have some concerns with the quality of experiments (3, 4 in __weaknesses__), technical novelty (5 in __weaknesses__), and reproducibility (6 in __weaknesses__).


**Strength And Weaknesses:**

##### Strengths
1. Development of efficient transformers is a challenging and important task. To my best knowledge, this is the first work on linear transformer specifically developed for graphs.
2. The proposed method seems to scale well to large datasets with up to 232k nodes.
##### Weaknesses
1. (Presentation of the method) The paper does not do a great job of presenting the proposed method and highlighting the technical novelty. Section 3.2 is not very accessible for readers not familiar with previous work on deformable transformers, and is hard to identify the technical contribution from the previous work. It seems that many new concepts introduced here such as sampling offsets or using interpolation to find the representation at fractional location have already been introduced in [1]. I recommend a separate background section including section 3.1 and more preliminaries of deformable transformers to make the paper self-contained.
2. (Clarity of the method) The main contribution of the work should be to use a NodeSort module to enable offset sampling and thus key node sampling. However, the presentation of this module is very short and there is even no detailed description about each sorting criteria, neither a reference provided. Besides, I am not sure some of them have a proper definition of ordering. For instance, if I understand correctly that BFS denotes the breadth-first search (the paper uses this abbreviation without giving the full name), then its order cannot be unique and could be exponential in the worst case. I expect more a more detailed presentation of all these sorting criteria. In addition to this, I also found some other missing details. For instance, how the offsets are computed is only described in the caption of Figure 1, which largely reduces the readability of the paper.
3. (Missing baselines) There are a few recent graph transformers leveraging better the structural information about the graph such as [2, 3]. In particular, [3] explore different linear/sparse attention methods from NLP, such as Performer or BigBird, to make graph transformers more scalable. I expect an empirical comparison with these methods to justify the practical impact of DGT.
4. (Weak results on ogbn-arxiv) A quick check on the leaderboard of [ogbn-arxiv](https://ogb.stanford.edu/docs/leader_nodeprop/#ogbn-arxiv) suggests that the SOTA performance is around 76% while the best performing methods in Table 1 of this work only achieves 71%, which is far behind the real SOTA. I am skeptical about the so-called SOTA performance on other datasets and the claim in the abstract that DGT achieves SOTA performance on 7 datasets.
5. (Novelty of positional encoding) The proposed Katz positional encoding is very similar to the random walk positional encoding introduced in [4, 5], which have already been widely used in many graph transformers. I recommend the authors to check and correct the claims on this contribution.
6. (Missing hyperparameter studies) Comparison of using different sorting criteria is shown in Table 3. However, the results are incomplete and results for only using PPR or Feat are not given. In addition, a study on the effect of different values of $\gamma$ would also be useful as its search space seems to be much larger than other hyperparameters.

##### Minor comments
1. The way of measuring FLOPs could be variable and ambiguous. Could the authors elaborate more on how they measured the FLOPs for different methods? Could the authors also report the runtime per epoch?
2. In section 3.3, "path" should be replaced with "walk" that allows repeated nodes.

##### References
[1] Zhu, Xizhou, et al. "Deformable DETR: Deformable Transformers for End-to-End Object Detection." ICLR 2020.

[2] Chen, Dexiong, et al. "Structure-aware transformer for graph representation learning." ICML 2022.

[3] Rampášek, Ladislav, et al. "Recipe for a General, Powerful, Scalable Graph Transformer." NeurIPS 2022.

[4] Dwivedi, Vijay Prakash, et al. "Graph Neural Networks with Learnable Structural and Positional Representations." ICLR 2021.

[5] Li, Pan, et al. "Distance encoding: Design provably more powerful neural networks for graph representation learning." NeurIPS 2020.

**Summary Of The Paper:**

Graph transformers have drawn growing attention in representation learning on graph-structured data. However, their success is limited to small graphs due to the quadratic complexity of the dot-product attention module in transformers. This paper proposes the Deformable Graph Transformer (DGT) to address this issue, by using a deformable graph attention (DGA) module with a linear complexity in the number of nodes. The proposed DGA, inspired by recent work in computer vision, only attend to a small set of key sampling nodes around a reference node. The set of key sampling nodes is learned by learning their positions in a sorted list pre-defined by each reference node based on some similarity criteria with respect to the reference node, such as personalized Pagerank score (PPR), BFS, or node feature similarity. The paper also proposes a positional encoding to use with the proposed DGT. The authors validate the proposed method on several node classification tasks.

**Summary Of The Review:**

Despite the efforts made by this work to solve a challenging and important for graph transformers, I recommend weak reject because 1) the presentation of method is imprecise and lacking a lot of details; 2) there are important missing baselines and the reported results seem to be far behind SOTA results; 3) the contribution to positional encoding is over-claimed due to incomplete related work; 4) some hyperparameter studies are missing.

---

> ### Author Response · Authors · 2022-11-18
> **Response to Reviewer wsk3 (2/2)**
>
> > Q4. Experimental results on ogbn-arxiv.
>
> We left out the frameworks, which require additional data, hand-crafted features, and self knowledge-distillation, etc. Although our method exhibits superior performance over strong baselines, we admit that there exist methods with higher accuracy on the leaderboard. We corrected our manuscript and toned down the expression ‘SOTA’.
>
> > Q5. Comparison of KatzPE with Random Walk Positional Encoding (RWPE)
>
> Our Katz PE is different from Random Walk Positional Encoding (RWPE) (Connectivity between two nodes under $k$-th hop v.s. Landing probability of a node to itself on $k$-th hop). We clarify the difference in the below:
>
> Random Walk Positional Encoding (RWPE) of $i$-th node is defined as
> $$p_i^{\text{RWPE}} = [RW_{ii}, RW_{ii}^2, \cdots, RW_{ii}^k] \in \mathbb{R}^k,$$
> where ${RW} = AD^{-1}$ is the random walk operator.
> So, RWPE expresses the landing probability of node $i$ to itself on $k$-th hop.
> On the other hand, our Katz PEs are defined as
> $$\text{KatzPE} (v_i) = MLP (\hat{A}[v_i]^T),$$
> where $\hat{\mathbf{A}} = \sum_{k=1}^K \beta^{k-1} \mathbf{A}^k$ and it represents **connectivity between nodes** with the decaying weight \beta to reflect the preference of path lengths. The equations clearly show that KatzPE does NOT have $D^{-1}$ and RW does NOT have $\beta^{k-1}$ and summation.
>
> For empirical comparison, we further report the performance gain of our KatzPE over RWPE below:
>
> |Positional Encoding| Chameleon | Cora | Citeseer | Squirrel |
> |-----|--|--|--|--|
> | RWPE |53.81 | 79.46 | 71.94 | 40.82
> | Katz PE (ours) | 72.13 | 81.44 | 73.02 | 59.79
>
> This result highlights that encoding connectivity between nodes is important for Transformers to understand graph structures.
>
> > Q6. Hyperparameter studies.
>
> We **already provided** experimental results for only using PPR or Feat i**n Section B.1 and Table 5**. As you mentioned, we also additionally provided the effect of different values of $\gamma$ on Cora dataset in the below table.
>
> |$\gamma$| 1 | 2| 4 | 8 | 16 |
> |---|--|--|--|--|--|
> |Acc|86.36|86.45|87.03|87.24|87.55|
>
> >Q7. More details on measuring FLOPs. Could the authors also report the runtime per epoch?
>
> We use flop counter provided by [https://github.com/facebookresearch/fvcore/blob/main/docs/flop_count.md](https://github.com/facebookresearch/fvcore/blob/main/docs/flop_count.md) to measure FLOPs.
> As you suggested, we additionally reported the runtime per epoch (s) on squirrel in the below table.
>
> |Method|Transformer|Graphormer|GT-full|GT-sparse|DGT-light|DGT|
> |--|--|--|--|--|--|--|
> |runtime (s)|0.59|0.50|0.50|0.27|0.27|0.36
>
> ----
> [1] Klicpera, Johannes, et al. "Predict then propagate: Graph neural networks meet personalized pagerank." ICLR 2019.
>
> [2] Chen, Dexiong, et al. "Structure-aware transformer for graph representation learning." ICML 2022.
>
> [3] Rampášek, Ladislav, et al. "Recipe for a General, Powerful, Scalable Graph Transformer." NeurIPS 2022.

---

> > ### Author Response · Authors · 2022-11-22
> > **Kind Reminder**
> >
> > Dear Reviewer wsk3,
> >
> > We sincerely appreciate your effort in reviewing our paper. We believe that your comment and feedback would be helpful for improving quality of our paper.
> >
> > During the discussion, we have responded to your comments and we believe that most of your suggestions have been resolved.
> >
> > Thank you again for your effort and time in reviewing our paper. Please let me know if you have further question or concerns.
> >
> > Best regards, Authors

---

> ### Author Response · Authors · 2022-11-18
> **Response to Reviewer wsk3 (1/2)**
>
> We sincerely thank you for your constructive comments for our paper. We address your concerns below.
>
> > Q1. Section 3.2 is not very accessible for readers not familiar with previous work on deformable transformers, and is hard to identify the technical contribution from the previous work. I recommend a separate background section including section 3.1 and more preliminaries of deformable transformers to make the paper self-contained.
>
> Thanks for the suggestion. We will provide more preliminaries of deformable attention proposed for 2D image processing in Section 3.1. We also provide more detailed comparison in the below:
>
> -   Discussion with Deformable DETR Our deformable graph attention addresses the challenges of applying deformable attention to graph-structured data. Since the deformable attention mechanism can only work on the regular space, it is **not directly applicable to graph data**. We address the challenge by transforming an irregular space into a set of regular spaces using our NodeSort module. Our NodeSort module is novel in the following aspects:
>     -   **Locality :** Deformable attention in Deformable DETR is defined on only one **global** grid whereas our deformable graph attention utilizes node sequences **locally** defined on each neighborhood (ego graph).
>     -   **Multiple relations :** Our Deformable Graph Attention (DGA) module uses diverse criteria (e.g. bfs, feature similarity, ppr) even in a neighborhood for constructing multiple node sequences to consider various relations in a graph.
>     -   **Semantic space :** We extended the deformable attention from a physical space to semantic space by performing NodeSort based on node feature similarity.
> -   We also improve learnability of offsets by replacing bilinear interpolation in deformable DETR with our kernel-based interpolation (Eq. (5)). Although this technique is simple, it largely improves the performance as shown in Table 6.
>
> > Q2.(a) More detailed presentations of NodeSort module
>
>
> Our NodeSort module converts a graph into a sorted sequence of nodes to embed nodes in the regular space without using learnable parameters. The criteria indicate how to sort our nodes. We use three criteria: BFS, Personalized PageRank~(PPR) score, and Feature similarity.
>
> -   Breadth First Search (BFS): Breadth First Search is a widely used algorithm for searching on graph-structured dataset. Given the graph and the base node, we set the base node as a root node and do breadth first search to get the sequence of nodes.
> -   Personalized PageRank score : Personalized PageRank score is a score that encodes the local neighborhood for root node. We use approximated personalized pagerank score [1] for NodeSort module. Given the graph and the base node, we calculate personalized page rank score as $\pi_{\text{ppr}}^{(k+1)} (i_x) = (1-\alpha)\hat{A}\pi_{\text{ppr}}^{(k)} (i_x) + \alpha \pi_{\text{ppr}}^{(k)}(i_x)$, where the x is the base node. Then, we sort nodes in descending order of personalized page rank score.
> -   Feature similarity : We use cosine similarity to compute feature similarity between nodes. Given the features of nodes and the base node, we measure the similarity score between feature of base node and other nodes through the cosine similarity. Then, we sort nodes in descending order of similarity score.
>
> BFS and PPR are used to generate node sequences based on structural similarity. In BFS, nodes with low shortest path distances from the base node are located in front of the sequence, and in PPR, nodes with high random walk scores from the base node are located in front of the sequence. Conversely, feature similarity is used for semantic proximity between nodes, and nodes with high semantic proximity to the base node are located in front of the sequence.
> By adopting diverse criteria for NodeSort module, our deformable graph attention simultaneously consider both semantic and structure proximity to design node sequences.
>
> > Q2. (b) Some missing details. For instance, how the offsets are computed is only described in the caption of Figure 1, which largely reduces the readability of the paper.
>
> We **already described** how the offsets are computed in 4-th paragraph of Section 3.2.
>
> > Q3. Empirical comparison with other baselines such as SAT [2] and GPS [3].
>
> Thank you for the suggestion. We additionally compare the performance of our DGT with other recent Transformer-based graph models such as SAT and GPS. The below table shows the performance comparison of graph Transformers and DGT on node classification tasks. The table shows that our DGT still outperforms other Transformer-based graph models.
>
> |Model| Chameleon | Cora | Citeseer | Squirrel |
> |-----|--|--|--|--|
> | SAT |55.75. | 81.84 |71.43 | 39.13 |
> | GPS-Bigbird | 56.43 | 79.84 | 72.44 | 39.31 |
> | GPS-Performer | 53.90 | 80.98 | 72.72 | 37.51 |
> | GPS-Transformer | 56.03 | 81.00 | 72.61 | 39.82 |
> | DGT (ours) |73.49 | 87.55 | 77.04 | 63.78 |

---

### Official Review · Reviewer_HE3Q · 2022-10-25

**Confidence:** 3
**Clarity, Quality, Novelty And Reproducibility:** quality, clarity and originality are …
**Correctness:** 4
**Technical Novelty And Significance:** 4
**Empirical Novelty And Significance:** 4
**Recommendation:** 5

**Strength And Weaknesses:**

Pros:
The motivation makes sense. The paper is easy to follow.

Cons:
1. The proposed method seems to be a graph sampling mechism + full attention on the sampled node subset. Therefore the claim of "sparse attention" should be clarified.
2. It's strange to run a Transformer-based graph model on full graph for node representation tasks without any graph sampling methods (e.g., random walk). Therefore, the comparision seems unfair to me from both the performance and the computation cost perspectives. More detailed benchmarking experiments with graph sampling methods will make the paper more solid.
3. The proposed Katz PE seems to be an important component contributing to the final performance, but the motivation of the Katz PE is less relevant to the main motivation of this paper. Therefore, it makes the solidness of the main contribution of this work (the graph samping mechnism) weak.

**Summary Of The Paper:**

This paper propose a new graph sampling mechanism for reducing the computational complexity in full attention of graph transformer. The sampling mechanism takes structural and semantic similarity into consideration. Experimental results show the effectiveness of the proposed method. Additionally proposed positioanl encoding enhances the performance of the model.

**Summary Of The Review:**

See Weaknesses. I'm willing to raise my score if my concerns are well addressed.

---

> ### Author Response · Authors · 2022-11-18
> **Response to Reviewer HE3Q**
>
> We sincerely thank you for your constructive comments for our paper. We address your concerns below.
>
> > Q1. The proposed method seems to be a graph sampling mechanism + full attention on the sampled node subset. Therefore the claim of "sparse attention" should be clarified.
>
> We’d like to clarify that we propose a learnable **sparse** (NOT full) attention (Deformable Graph Attention) that can aggregate only some key nodes rather than all key nodes, for each query node. In other words, our model does not sample the input graph in advance but learns to sparsely give attention scores to only some useful nodes among all nodes. Specifically, unlike the previous full attention mechanism, our model predicts multiple offsets for each query node and interpolates the neighbors of the nodes where the offsets point, and finally aggregates the interpolated nodes to represent the query node. This offset-based deformable attention allows us to circumvent the pairwise operation in full attention mechanisms.
>
> > Q2. It's strange to run a Transformer-based graph model on full graph for node representation tasks without any graph sampling methods (e.g., random walk). Therefore, the comparison seems unfair to me from both the performance and the computation cost perspectives. More detailed benchmarking experiments with graph sampling methods will make the paper more solid.
>
> As mentioned in the first point, we did not proceed with the graph sampling method. Therefore, we fairly compared our sparse attention with the conventional full attention methods under the same conditions without graph sampling. Similar to the graph sampling you mentioned, we compared our model with GT-sparse which full attention was applied after sampling only the 1-hop connectivity relationship of the graph.
>
> > Q3. The proposed Katz PE seems to be an important component contributing to the final performance, but the motivation of the Katz PE is less relevant to the main motivation of this paper. Therefore, it makes the solidness of the main contribution of this work (the graph samping mechnism) weak.
>
> Our main contribution is a learnable sparse Transformer to improve the scalability as well as the expressive power of the **Transformer**-based graph models. Further, in the same direction to effectively design the Transformer to the graph, we propose Positional encoding, KatzPE, since designing the positional encoding in **Transformer** is a significant issue and the positional encoding for Transformer-based graph models has been less studied.

---

> > ### Author Response · Authors · 2022-11-22
> > **Kind Reminder**
> >
> > Dear Reviewer HE3Q,
> >
> > We sincerely appreciate your effort in reviewing our paper. We believe that your comment and feedback would be helpful for improving quality of our paper.
> >
> > During the discussion, we have responded to your comments and we believe that most of your suggestions have been resolved.
> >
> > Thank you again for your effort and time in reviewing our paper. Please let me know if you have further question or concerns.
> >
> > Best regards, Authors

---

> > ### Author Response · Authors · 2022-11-22
> > **Update in [Q2]**
> >
> > We additionally conduct the experimental results in comparison of our DGT with Transformer-based graph models with 1-hop and 2-hop connectivity in the below table. As shown in the table, our DGT outperforms Transformer-based graph models with 1-hop and 2-hop connectivity. This indicates that selecting informative nodes to aggregate is important for learning node representations compared to Transformer with naive graph sampling method.
> >
> > | |Chameleon | Cora | Citeseer | Squirrel
> > |--|:--:|:--:|:--:|:--:|
> > 1-hop |64.82 | 85.63 | 75.49 | 44.22
> > 2-hop |64.88 | 84.61 | 74.04 | 54.67
> > DGT.  |73.48|87.55|77.04|63.78

---

### Official Review · Reviewer_PKKU · 2022-10-26

**Confidence:** 4
**Correctness:** 3
**Technical Novelty And Significance:** 2
**Empirical Novelty And Significance:** 3
**Recommendation:** 5

**Clarity, Quality, Novelty And Reproducibility:**

For clarity, some details are missing about deformable graph attention module.
For quality, the writing is easy to follow.
For novelty, the novelty and technical contribution are limited.
For reproducibility, the code is attached.



**Strength And Weaknesses:**

Strengths
1. The research problem is a fundamental and crucial one.
2. The experimental results show superior accuracies and efficiency (FLOPs).

Weaknesses
1. The novelty and technical contribution are limited.
2. It is unclear for the deformable graph attention module.
3. It is unclear why the proposed method has lower computational complexity.

Detailed comments:
1. What is the motivation to choose personalized pagerank score, bfs, and feature similarity as sorting criteria?
2. For NodeSort, 1) how to choose the base node, or is every node a base node?
2)“NodeSort differentially sorts nodes depending on the base node.” Does this mean that the base node affects the ordering,  affects the key nodes for attention, and further affects the model performance?
3)After getting the sorted node sequence, how to sample the key nodes for each node? And how many key nodes are sampled？is the number of key nodes a hyper-parameter? 4)What are the Value nodes used in Transformer in this paper? 5)How to fuse node representations generated by attention for different ranking criteria.
3. Intuitively, the design of deformable graph attention is complicated, and the Katz positional encoding involves the exponentiation of adjacency matrix, so Is the computational complexity really reduced? Where can the reduction in complexity be explained from the proposed method compared to baselines? or just from the sparse implementation?

**Summary Of The Paper:**

This paper proposes deformable graph transformer (DGT) to efficiently perform attention on graphs. The deformable graph transformer mainly consists of two components: deformable graph attention and Katz positional encoding. Experiments on eight datasets show better performance.

**Summary Of The Review:**

The deformable graph attention module should be presented in a clear way. The reduction of complexity should be explained further.

---

> ### Author Response · Authors · 2022-11-18
> **Response to Reviewer PKKU**
>
> We sincerely thank you for your constructive comments for our paper. We address your concerns below.
>
> > Q1. The novelty and technical contribution of proposed methods.
>
> To the best of our knowledge, our DGT is the first work to propose **learnable sparse Transformer** for dealing with **large-scale graphs**. In other words, we modify the full self-attention module used in Transformer into our deformable attention module. On the other hand, most existing works mainly focus on modulating the input of Transformer or designing positional encoding to inject graph structural information. Our deformable graph attention can be combined with existing works.
>
> > Q2. Motivation to choose personalized page rank score, bfs, and feature similarity as sorting criteria.
>
> BFS and PPR are used to generate node sequences based on structural similarity. In BFS, nodes with low shortest path distances from the base node are located in front of the sequence, and in PPR, nodes with high random walk scores from the base node are located in front of the sequence. Conversely, feature similarity is used for semantic proximity between nodes, and nodes with high semantic proximity to the base node are located in front of the sequence.
> By adopting diverse criteria for NodeSort module, our deformable graph attention simultaneously consider both semantic and structure proximity to design node sequences.
>
> > Q3. Clarity of Deformable Graph Attention
>
> - Q3.1) How to choose the base node?
>
> From Eq. (4), query node is selected as the base node of sorted sequences $S_{\pi q}$.
>
> - Q3.2) “NodeSort differentially sorts nodes depending on the base node.” Does this mean that the base node affects the ordering, the key nodes for attention, and further affects the model performance?
>
> Yes, the base node affects the ordering and the model performance. From Table 3, absolute ordering which uses same base node for all query nodes, underperforms relative ordering, which uses query nodes as a base node. It indicates that selecting base node is important for the performance.
>
> - Q3.3) How to sample the key nodes for each node and how many key nodes are sampled?
>
> We use $\tilde S_{\pi q}(\Delta(\mathbf p))$ for keys, which are computed by the kernel-based interpolation with nodes selected based on offset $\Delta (\mathbf p)$ as in Section 4.2.
> Specifically, we first compute offset $\Delta (\mathbf p) = \theta_{\pi m k}^{\text{off}} (\mathbf z_q )$, where $\theta^{\text{off}}$ is a linear function with an activation function, for each query node $v_q$. Then, we construct $K$ number of Key representations $\tilde S_{\pi q}(\Delta \mathbf p)$ via kernel-based interpolation with $2\epsilon$ number of nodes as formulated in Eq (5). Finally, we aggregate Keys with the attention module defined in Eq. (4).
>
> - Q3.4) What are the Value nodes used in Transformer in this paper?
>
> Same as self-attention in a standard Transformer, value representation is the same as key representation. So, we use representations of each value (key) as $\tilde S_q(\Delta \mathbf{p}_{\pi mqk})$.
>
> - Q3.5) How to fuse node representations generated by attention for different criteria?
>
> As shown in Eq.(4), the node representations for different criteria are aggregated by summation following linear projection.
>
> > Q4. Discussion on Complexity compared to baselines.
>
> We **already discussed** why our deformable graph attention is efficiently better than the self-attention in most Transformer-based graph networks **in Section 3.5 and Section A**. Computing self-attention requires $\mathcal{O}\left(N^2 C + N C^2\right)$, which is **a quadratic complexity** with respect to the number of nodes N. The complexity of deformable graph attention (DGA) is $\mathcal{O}\left(N C^2 T + NWKCT \right)$, which is **a linear complexity** with respect to the number of nodes $N \gg W,K,C,T$. In the case of Katz PE, following other Transformer-based graph models, we compute $\hat{A}$ in pre-processing step.

---

> > ### Author Response · Authors · 2022-11-22
> > **Kind Reminder**
> >
> > Dear Reviewer PKKU,
> >
> > We sincerely appreciate your effort in reviewing our paper. We believe that your comment and feedback would be helpful for improving quality of our paper.
> >
> > During the discussion, we have responded to your comments and we believe that most of your suggestions have been resolved.
> >
> > Thank you again for your effort and time in reviewing our paper. Please let me know if you have further question or concerns.
> >
> > Best regards, Authors

---

### Official Review · Reviewer_Xtmm · 2022-10-31

**Confidence:** 3
**Correctness:** 3
**Technical Novelty And Significance:** 3
**Empirical Novelty And Significance:** 2
**Recommendation:** 5

**Clarity, Quality, Novelty And Reproducibility:**

- Clarity: I found the paper extremely hard to read. There are many ambiguities, the notation is unnecessarily complex, and many parts need to be specified more clearly.
- Novelty: The idea of ordering and subsampling the nodes is novel and interesting.
- Reproducibility: Because the preprocessing is unspecified, I would probably not be able to reproduce the paper.
- Quality: The quality of the paper is not enough to recommend acceptance.



**Strength And Weaknesses:**

Strengths:
 - better performance on various datasets
 - the topology-based sorting ideas are interesting and novel to the best of my knowledge

 Weaknesses:
  - some parts (for example, the preprocessing) are not specified in the paper
  - the paper is difficult to read
  - some design choices need more justification
  - very complex
  - claiming that this method is the best among the transformer-based ones is somewhat unfair. The main difference between the transformer-based methods and the GAT-style GNNs is the unrestricted attention, but here the preprocessing step introduces very similar restrictions.

The main weakness of the method:
- From Table 3, picking nodes at random is not much worse than choosing the nodes in the smart way proposed by the paper. This is surprising and questions what deformable attention does and its importance.


**Summary Of The Paper:**

The authors propose a new transformer-like architecture for processing graphs. The main component is a special sparse attention module that considers only the neighbor nodes based on certain metrics. These metrics can be either structure or feature-based. For sampling from the neighborhood, it uses learned offsets. The authors also propose a novel position encoding method for the nodes in the graph.


**Summary Of The Review:**


To recommend acceptance, I would like to see the following:
- Clarity has to be improved
- Specification of the preprocessing step in a clear way, including in the complexity calculation
- I would like to see an analysis of what the KatzPE does
- Some analysis on why the DGA's performance is very close to random


Questions to clarify:
- How do the sorting criteria work? What are they exactly? What is their input? Do they have learnable parameters?
- The paper claims that it is more efficient than the transformer-based methods. However, it does not consider the preprocessing step, which can be very slow (although it has to be done only once). Also,  "feature similarity", as far as the name suggests, is just ordinary attention, so it should be the same speed.
- "Although restricting the attention scope to local neighbors is a simple remedy to reduce the computational complexity, it leads to a failure in capturing local-range dependency, which is crucial for large-scale or heterophilic graphs." -> do you have any reference for this? According to Table 3, even using random nodes is almost as good as the best model. Isn't BFS doing just that?
- You mention that "DGT-light" uses a single sorting criterion. Which one?
- All of Eq. (4) is position based, but the positions are defined by the most relevant features (by S). The attention matrix only depends on the query node's feature and not the key's (it is just a fixed function for a given key index, k). This can act as a regularizer because the network can hardly choose individual, well-defined nodes to attend to. This can by itself improve performance.
- What is p? It can be inferred that it's probably an offset vector, but it would be nice if that is written in the paper.
- It would be nice to visualize p for some learned tasks. It would be informative about how far the model attends.
- What value of \gamma is used in eq (5)?
- "A major issue with positional encoding on graphs is the absence of absolute positions of nodes, unlike other domains." - what does it mean to have absolute positional encodings in a graph? What defines the ordering of the nodes?
- I don't see why KatzPE makes sense. It would be nice if the authors could add some more explanation. For me, it seems more like some kind of structure descriptor added to the nodes than it has anything to do with positions.
- A_k in Eq. (6) is of a fixed size. Does this mean that, unlike the transformer-based methods, KatzPE depends on the graph size? (for big graphs, this limitation is not present since there are only N' anchor nodes)
- It would be worth writing down that the so-called sampling is not actually an active sampling operation but just taking the features from the learned offsets. It can be confusing on the first read.
- Could you provide more details on how O(NC^2T + NKC^2T + WNKCT) becomes O(NC^2T + WNKCT) in Appendix A? Why does swapping the interpolation help with this?
- In 4.4, Figure 2, a should be Figure 2a
- In Appendix C.2, the list of parameters should be a table

---

> ### Author Response · Authors · 2022-11-18
> **Response to Reviewer Xtmm (2/2)**
>
> > Q6. The attention matrix only depends on the query node's feature and not the key's.
>
> Since the input nodes of deformable graph attention have their order on the sequences, attention matrix can be computed without pair-wise multiplication between Query and Key. There has been some recent works [4] that computes the attention in this manner.
>
> > Q7. Details of “p”. What is p? It can be inferred that it's probably an offset vector, but it would be nice if that is written in the paper.
>
> $\mathbf{p}$ is the position(or index) on the sorted list. For example, if $\mathbf{p}$ is set to 5, then $S(p)$ indicates features of 5-th node on the sequence $S$.
> We clarified it in our revised version.
>
> > Q8. It would be nice to visualize p for some learned tasks. It would be informative about how far the model attends.
>
> Good suggestion. As you suggested, we visualize $\mathbf{p}$ in Appendix of our revised version. The visualization shows that the model can capture long-range dependencies.
>
> > Q9. What value of $\gamma$ is used in Eq (5)?
>
> In the supplement, we already provided the search space of $\gamma$.
>
> > Q10. “A major issue with positional encoding on graphs is the absence of absolute positions of nodes, unlike other domains” -> what does it mean to have absolute positional encodings in a graph?
>
> First, we did not mention about “absolute positional encoding”. The referred sentence means that graph-structured data are in an irregular space unlike 2D image data, which are in the Euclidean space.
>
> > Q11. KatzPE seems more like some kind of structure descriptor than it has anything to do with positions.
>
> The Katz index represents the number of K hop or shorter paths between nodes. The higher Katz index means stronger connection between nodes on graphs. Thanks to decaying factor $\beta$, longer paths are properly discounted. So, two nodes with shorter paths have a higher Katz index. In other words, Katz index captures a relative position as well. Likewise, the eigenvector of graph laplacian, which is the most well-known positional encoding in graphs, is also a positional encoding derived from an adjacency matrix. This is Fourier basis on a graph that captures local structure as well as relative distance between nodes.
>
> > Q12. Unlike the transformer-based methods, KatzPE depends on the graph size?
>
> Many positional encodings for transformer-based graph methods are derived from an adjacency matrix, which means that they are dependent on the graph size. For example, positional encodings based on eigenvectors of graph laplacian are dependent on the graph size when computing eigendecomposition. Similar to our approach, for efficient computations, various approximations have been studied in the literature.
>
> > Q13. It would be worth writing down that the so-called sampling is not actually an active sampling.
>
> Thank you for your helpful comment. We followed the terminology of Deformable DETR for sampling but as suggested, we changed “sampling” to “selecting” to avoid the confusion in the revised version.
>
> > Q14. Details on how $\mathcal{O}(NC^2T + NKC^2T + WNKCT)$ becomes $\mathcal{O}(NC^2T + WNKCT)$
>
> We can calculate the linear transformation of the interpolated features $\mathbf W_{\pi m}' \tilde S_{\pi q} (\Delta \mathbf p_{\pi m q k})$ by interpolation of the linear transformed features $\mathbf{W}'_{\pi m}\mathbf{X}$. So, we do not have a linear projection for each key. Thus, the complexity of $NC^T + NKC^T +WNKCT$ can be $NC^T + NC^T +WNKCT$. Then, the complexity of calculating Eq. (4) becomes $\mathcal{O}(NC^T +WNKCT)$.
>
> > Q15. In 4.4, Figure 2, a should be Figure 2a.
>
> Thank you for your correction. We corrected it in our revised version.
>
> > Q16. In Appendix C.2, the list of parameters should be a table.
>
> Thank you for your suggestion. We updated it accordingly in our revised version.
>
> ____
> [1] Klicpera, Johannes, et al. "Predict then propagate: Graph neural networks meet personalized pagerank." ICLR 2019.
>
> [2] Liu Meng, et al. "Non-local graph neural networks." TPAMI 2021.
>
> [3] Hongbin Pei, et al. "Geom-GCN: Geometric Graph Convolutional Networks." ICLR 2020.
>
> [4] Zhu, Xizhou, et al. "Deformable detr: Deformable transformers for end-to-end object detection." ICLR 2021.

---

> ### Author Response · Authors · 2022-11-18
> **Response to Reviewer Xtmm (1/2)**
>
> We sincerely thank you for your constructive comments for our paper. We address your concerns below.
>
> > Q1. Claiming that this method is the best  among the transformer-based ones is somewhat unfair. The main difference between the transformer-based methods and the GAT-style GNN is the unrestricted attention, but here the processing step introduces very similar restrictions.
>
> We respectfully disagree with your opinion. We **already compared** our model with GT-sparse which full attention was applied after sampling only the 1-hop connectivity relationship of the graph **like GAT-style GNN** in **Table 1**. Compared to GT-sparse, our DGT shows better performance, which indicates that adaptively selecting informative node is important.
>
> > Q2. From Table 3, even using random nodes is almost as good as the best model.
>
> Table 3 was misread by Reviewer Xtmm . From Table 3, the best model (the relative ordering with multiple criteria) shows a performance improvement up to 6.02(%) and 3.78(%) on average compared to the model with random permutation. Also, as mentioned on ‘NodeSort module’ paragraph in Section 4.3, the reason why the absolute ordering with BFS shows no performance compared to randomly permuted absolute ordering is because a single absolute node sequence is not sufficient to learn complex relationships between nodes in the graph.
>
> > Q3. More details on criteria used in NodeSort Module.
>
> Our NodeSort module converts a graph into a sorted sequence of nodes to embed nodes in the regular space without using learnable parameters. The criteria indicate how to sort our nodes. We use three criteria: BFS, Personalized PageRank~(PPR) score, and Feature similarity.
>
> -   Breadth First Search (BFS): Breadth First Search is a widely used algorithm for searching on graph-structured dataset. Given the graph and the base node, we set the base node as a root node and do breadth first search to get the sequence of nodes.
> -   Personalized PageRank score : Personalized PageRank score is a score that encodes the local neighborhood for root node. We use approximated personalized pagerank score [1] for NodeSort module. Given the graph and the base node, we calculate personalized page rank score as $\pi_{\text{ppr}}^{(k+1)} (i_x) = (1-\alpha)\hat{A}\pi_{\text{ppr}}^{(k)} (i_x) + \alpha \pi_{\text{ppr}}^{(k)}(i_x)$, where the x is the base node. Then, we sort nodes in descending order of personalized page rank score.
> -   Feature similarity : We use cosine similarity to compute feature similarity between nodes. Given the features of nodes and the base node, we measure the similarity score between feature of base node and other nodes through the cosine similarity. Then, we sort nodes in descending order of similarity score.
>
> BFS and PPR are used to generate node sequences based on structural similarity. In BFS, nodes with low shortest path distances from the base node are located in front of the sequence, and in PPR, nodes with high random walk scores from the base node are located in front of the sequence. Conversely, feature similarity is used for semantic proximity between nodes, and nodes with high semantic proximity to the base node are located in front of the sequence.
> By adopting diverse criteria for NodeSort module, our deformable graph attention simultaneously consider both semantic and structure proximity to design node sequences.
>
>
> > Q4. Complexity of the preprocessing step.
>
> Multiple sorting methods are computed **in the pre-processing and done only once** similar to positional encodings in recent Transformer-based graph models that compute eigendecomposition.
>
> > Q5. Reference for the claim that “Although restricting the attention scope to local neighbors is a simple remedy to reduce the computational complexity, it leads to a failure in capturing local-range dependency, which is crucial for large-scale or heterophilic graphs."
>
> Several papers [2,3] claimed that the message-passing schemes lack the ability to capture long-range dependencies in heterophilic graphs. Restricting the attention scope to neighbors can be considered as the message passing scheme with the attention mechanism. So, restricting the attention scope to neighbors leads to a failure in capturing long-range dependency, which is crucial for large-scale or heterophilic graphs.
>
>
> > Q5. You mention that "DGT-light" uses a single sorting criterion. Which one?
>
> For DGT-light, we use BFS for the criterion.

---

> > ### Comment · Reviewer_Xtmm · 2022-11-18
> > **Thank you for your response**
> >
> > Thanks to the authors for their detailed response. Unfortunately, I don’t see any updates to the paper, but according to the authors' comments, they will be made in the future. I still have concerns about reproducibility and clarity. Also, I think that the improvements from DGA are modest compared to its complexity. I’m increasing my score because of the clarifications of the authors.
> >
> > > We respectfully disagree with your opinion. We already compared our model with GT-sparse which full attention was applied after sampling only the 1-hop connectivity relationship of the graph like GAT-style GNN in Table 1. Compared to GT-sparse, our DGT shows better performance, which indicates that adaptively selecting informative node is important.
> >
> > I have not claimed that they became equivalent, but it is similar in restricting attention to a few nodes, which are probably the closeby ones. Of course, it is impossible to tell what the network attends to without seeing an analysis of the sampling offsets. Perhaps a GAT-style GNN with 2-hop connectivity would have a similar performance.
> >
> > That said, it is also a nice property: the proposed model is in-between the fully restricted GAT and the unrestricted transformer models.
> >
> > > Table 3 was misread by Reviewer Xtmm . From Table 3, the best model (the relative ordering with multiple criteria) shows a performance improvement up to 6.02(%) and 3.78(%) on average compared to the model with random permutation
> >
> > I have noticed this difference. However, there is only a 3.78% average improvement compared to random ordering, which was quite surprising.
> >
> > > Feature similarity : We use cosine similarity to compute feature similarity between nodes. Given the features of nodes and the base node, we measure the similarity score between features of base node and other nodes through the cosine similarity. Then, we sort nodes in descending order of similarity score.
> >
> > What are exactly the features of the nodes? The raw features without any preprocessing? Or an embedding that is learned from those (in this case, I don’t see how the embeddings are trained)?
> >
> > The Personalized PageRank and BFS seem to be qualitatively similar, but the paper has no ablations showing whether both are necessary (Table 3). Do you have any comments on this?
> >
> > > Multiple sorting methods are computed in the pre-processing and done only once similar to positional encodings in recent Transformer-based graph models that compute eigendecomposition.
> >
> > Well, the preprocessing step also contributes to running the algorithm. It's unfair to move certain operations as preprocessing and claim that the resulting algorithm runs faster just because the preprocessing step is not considered in the complexity metric. Exponentiating A seems to be especially costly, but doing N sorts (one sort per node) is also non-negligible.

---

> > > ### Author Response · Authors · 2022-11-22
> > > **Re: Thank you for your response**
> > >
> > > Thank you for your response, we sincerely appreciate your constructive comments. We respond to your comments in the below:
> > >
> > > > Of course, it is impossible to tell what the network attends to without seeing an analysis of the sampling offsets. Perhaps a GAT-style GNN with 2-hop connectivity would have a similar performance. That said, it is also a nice property: the proposed model is in-between the fully restricted GAT and the unrestricted transformer models.
> > >
> > > From Figure 2 (b), our DGT aggregates messages from remote nodes beyond 1-hop neighbors through the attention mechanism. Also, we observed that the ratio of the attention score for the nodes with the same label is high (0.97 as mentioned in Section 4.4), which means that our DGT effectively performs the sparse attention.
> > >
> > > Moreover, we report the ratio of the number of nodes to attend according to the distance ($k$) between the given node $v_i$ and the other nodes on Chameleon dataset in the below table. The table shows that our DGT aggregates information from both local and remote nodes.
> > >
> > > |$k$|1|2|3|<3|
> > > |--|--|--|--|--|
> > > |Ratio|0.62|0.31|0.05|0.02|
> > >
> > > As Reviewer Xtmm mentioned, we also compare our DGT with Transformer-based graph models with 2-hop connectivity in the below table. From the table, our DGT outperforms the model with 2-hop connectivity as well as the model with 1-hop connectivity. This indicates that selecting informative nodes to aggregate is important for learning node representations.
> > >
> > > | |Chameleon | Cora | Citeseer | Squirrel|
> > > |--|:--:|:--:|:--:|:--:|
> > > |1-hop |64.82 | 85.63 | 75.49 | 44.22|
> > > |2-hop |64.88 | 84.61 | 74.04 | 54.67|
> > > |DGT  |73.48|87.55|77.04|63.78|
> > >
> > > > I have noticed this difference. However, there is only a 3.78% average improvement compared to random ordering, which was quite surprising.
> > >
> > > These results indicate that our deformable graph attention is robust on the sorting with diverse criteria.
> > >
> > > > What are exactly the features of the nodes? The raw features without any preprocessing? Or an embedding that is learned from those (in this case, I don’t see how the embeddings are trained)?
> > >
> > > We use the raw feature without any preprocessing for calculating feature similarity.
> > >
> > > > The Personalized PageRank and BFS seem to be qualitatively similar, but the paper has no ablations showing whether both are necessary (Table 3). Do you have any comments on this?
> > >
> > > In BFS, nodes with low shortest path distances from the base node are located in front of the sequence, and in PPR, nodes with high random walk scores from the base node are located in front of the sequence. So they are slightly different. Also, we conducted ablation studies of PPR and BFS in the below table. The table shows that using both BFS and PPR improves performance over the models using each of them alone.
> > >
> > > |Criteria | Chameleon | Cora | Citeseer | Squirrel|
> > > |--|:--:|:--:|:--:|:--:|
> > > |BFS |73.04 | 86.60 | 75.72 | 62.58|
> > > |PPR |72.66 | 86.31 | 75.42 | 62.57|
> > > |BFS,PPR | 72.78 | 87.04 |76.22 | 62.96|
> > > |BFS,PPR,Feat | 73.48 |87.55 |77.04 | 63.78 |
> > >
> > > > Well, the preprocessing step also contributes to running the algorithm. It's unfair to move certain operations as preprocessing and claim that the resulting algorithm runs faster just because the preprocessing step is not considered in the complexity metric. Exponentiating A seems to be especially costly, but doing N sorts (one sort per node) is also non-negligible.
> > >
> > > We agree that the preprocessing step also contributes to the running time of algorithms. So, we compare the running time of our node sorting (e.g., BFS, PPR score, and feature similarity) with Laplacian eigenvalue decomposition (Laplacian EigVec.) used in most Transformer-based graph models in the below table. Our node sorting schemes are efficient and faster than Laplacian EigVec. Compared to the overall training time (~300s), our preprocessing time (< 1s) is negligible.
> > >
> > > |  |Chameleon| Cora| Citeseer | Squirrel|
> > > |--|:--:|:--:|:--:|:--:|
> > > Laplacian EigVec|0.72s | 0.91s |1.80s | 6.01s|
> > > BFS | 0.40s |0.46s | 0.48s | 0.55s|
> > > PPR | 0.27s |0.29s | 0.31s | 0.32s|
> > > Feat sim. | 0.31s | 0.27s | 0.26s | 0.27s|

---

### Author Response · Authors · 2022-11-18
**Summary of updates in the revision.**


We sincerely thank all reviewers for constructive comments. We believe that they have contributed to improving the quality of our paper a lot.  We have revised our paper based on all reviewer’s suggestions. The major changes are summarized as below:



**1. More details of NodeSort Module. (Reviewer Xtmm, PKKU, wsk3)**

-   We have **clarified details of NodeSort module** in the Appendix.

**2. Experimental results on other tasks. (Reviewer KagM)**

-   We **have compared our DGT on graph regression task** with ZINC dataset in the Appendix.

**3. Discussion of Katz PE (Reviewer  Xtmm, HE3Q, wsk3, KagM)**

-   We **have provided the discussion of Katz PE** in the Appendix.
-   We **have compared KatzPE with a RWPE** according to the suggestion in Section  4.3 and Table 4.

**4. More preliminaries for readability. (Reviewer wsk3)**
- We **have included more preliminaries** of deformable attention proposed for 2D image processing in Section 3.1
- We  have also **provided more detailed comparisons** of DGT with deformable attention in Appendix.

---

### Author Response · Authors · 2022-12-07
**The end of discussion phase is approaching.**

Dear Reviewers,

Thank you again for your effort in reviewing our paper. Since we have interactions with you only by this Monday (12nd), please let me know if you have further questions or concerns. We look forward to your feedback and discussion.

Best regards, Authors

---

### Decision · Program_Chairs · 2023-01-20

**Decision:**

Reject

**Justification For Why Not Higher Score:**

During the AC-Reviewer discussion, all Reviewers reached a consensus to recommend a rejection for this work.

**Justification For Why Not Lower Score:**

N/A

**Metareview: Summary, Strengths And Weaknesses:**

In this work, a deformable graph transformer is proposed, which is capable of sparsifying the attention mechanism to make Transformers more scalable. Additionally, a learnable Katz PE is proposed in order to improve the representational quality of the model. Reviewers have all agreed that the work has relevance for the community, and the proposed solution is interesting. However, serious concerns were raised by the motivation of the proposed combination of methods, their practical impact, the novelty of the various components, and certain non-solid claims made by the authors. After a thorough AC-Reviewer discussion (see details), a consensus was reached to reject this paper, which I agree with. I hope the authors can take all the reviewer comments into account, and continue this line of work -- it could be quite promising!

**Summary Of Ac-Reviewer Meeting:**

First, a disclaimer -- not all of the reviewers were present for the meeting due to scheduling / time-zone constraints. That being said, the meeting did have a diversity of opinions, featuring the sole Reviewer with the positive score, and the other Reviewers communicated their thoughts via email before the meeting took place, so we were able to discuss their opinions as well.

The reviewers highlighted the following reasons about accepting the work:

* The problems considered, such as sparsifying graph Transformers, are very important.
* The reported results appear reasonable, and favourable for the proposed method.
* Sorting the nodes in the attention was perceived to be novel.
* The rebuttal provided a graph-level task---which might benefit more from Transformers---while the original paper only had node-level problems.

The downsides, on the other hand, included:

* The explanation and motivation of the deformable mechanism is unclear, and it was not necessarily improved after the rebuttal. A lack of clarity was also noted around the node sorting mechanism.
* There is a repeated emphasis on the need to select 'similar' nodes for the attention mechanism. It has been argued by the reviewers that, in fact, attention may also want to consider dissimilar nodes, depending on the task, which would limit the method's broader applicability.
* The positional encoding scheme feels a lot like (a) diffusion-based methods, and (b) distance-encoding GNNs, and hence concerns were raised about its novelty.
* There is a notable lack of comparisons to previous sparse transformers proposed in the literature, most notably GraphGPS (Rampášek et al., NeurIPS'22).
* It is unclear why FLOPS are the best measure of efficiency in this case.

My opinion, which the reviewers concurred with in the call, is that the cons significantly outweigh the pros in this case. Specifically:

* Both the idea of sparsifying Transformers [see GraphGPS; Rampášek et al., NeurIPS'22], and the idea of sorting nodes by attention coefficients [see GOAT; Chatzianastasis et al., AAAI'23], are not novel. I am willing to not take it strongly into account that the authors did not cite or compare against GOAT, given their publication time relative to the ICLR deadline, and the fact none of the reviewers raised this initially in their reviews.
* That being said, still, the authors propose a fairly contrived-looking method for obtaining sparsity, and therefore it is vital to compare against other sparse Transformer schemes (with the exception of GAT and GT-sparse, which use the original graph). In their rebuttal, the authors provide comparisons to GraphGPS. The reviewer raising this issue was not convinced that the GraphGPS parameters were properly tuned for the node-level tasks the authors presented. And for the graph-level task (ZINC), where the short-cutting effect of Transformers is expected to be most pronounced, the proposed method loses out _significantly_ to the reported GraphGPS results. Further, on ZINC the proposed method is worse than the Graphormer (a dense Transformer) as well.
* All in all, I am unconvinced that this method is an empirically stronger option than GraphGPS based on the experiments presented. More work is needed to ascertain that conclusion, and more discussion is needed to compare the two methods.
* While the above points may not necessarily be enough to select the paper for rejection, it is absolutely paramount that the proposed mechanism and its perceived utility are explained well. It seems clear that Reviewers, by majority, did not completely understand _why_ the proposed mechanism should be used in the first place. Further, the Authors' rebuttal did not improve the Reviewers' confidence. This, to me, is yet another important 'red flag' about accepting the work in its current form.

I hope the authors take into account this feedback (as well as all the Reviewers' comments) when improving their work further.